# Impact of intensifying nitrogen limitation on ocean net primary production is fingerprinted by nitrogen isotopes

Pearse J. Buchanan [1✉], Olivier Aumont[2], Laurent Bopp [3], Claire Mahaffey [1] & Alessandro Tagliabue [1]

The open ocean nitrogen cycle is being altered by increases in anthropogenic atmospheric nitrogen deposition and climate change. How the nitrogen cycle responds will determine long-term trends in net primary production (NPP) in the nitrogen-limited low latitude ocean, but is poorly constrained by uncertainty in how the source-sink balance will evolve. Here we show that intensifying nitrogen limitation of phytoplankton, associated with near-term reductions in NPP, causes detectable declines in nitrogen isotopes ($\delta^{15}$N) and constitutes the primary perturbation of the 21$^{st}$ century nitrogen cycle. Model experiments show that ~75% of the low latitude twilight zone develops anomalously low $\delta^{15}$N by 2060, predominantly due to the effects of climate change that alter ocean circulation, with implications for the nitrogen source-sink balance. Our results highlight that $\delta^{15}$N changes in the low latitude twilight zone may provide a useful constraint on emerging changes to nitrogen limitation and NPP over the 21$^{st}$ century.

[1] Department of Earth, Ocean and Ecological Sciences, University of Liverpool, Liverpool, UK. [2] Laboratoire d'Océanographie et du Climat: Expérimentations et Approches Numériques (LOCEAN), IPSL, Sorbonne Université, IRD, CNRS, MNHN, Paris, France. [3] Laboratoire de Météorologie Dynamique (LMD), IPSL, Ecole Normale Supérieure - Université PSL, Sorbonne Université, Ecole Polytechnique, CNRS, Paris, France. ✉email: pearse.buchanan@liverpool.ac.uk

Nitrogen limits phytoplankton growth in the low latitudes (45°S to 45°N), making the oceanic nitrogen cycle an important regulator of net primary production (NPP), which is essential for ecosystem health and carbon sequestration[1,2]. Two major anthropogenic drivers, atmospheric nitrogen deposition and climate change, are altering the open-ocean nitrogen cycle[3]. Atmospheric nitrogen deposition to the ocean has doubled since the preindustrial era[4,5] and is currently the main input of anthropogenic nitrogen to the open ocean[6], causing documented increases in nitrate in certain locations[7]. It is greatest in the Northern Hemisphere and is expected to peak by year 2030 as clean air initiatives are implemented[5,8]. Meanwhile, climate change is accelerating[9] and, by altering the properties and circulation of seawater[10–12], modifies biogeochemical processes and the distribution of bioavailable nitrogen[3]. These anthropogenic drivers threaten to alter the twenty-first century nitrogen cycle and the nature of this alteration could either mitigate or amplify projected marine ecosystem productivity decline in the coming centuries[13].

Earth System Models (ESMs) are used to project anthropogenic trends[10], identify their drivers[14] and assess their detectability using concepts like the Time of Emergence (ToE), which quantify when anthropogenic signals emerge from natural variability[15]. Although multi-century projections herald massive NPP declines associated with increasing nutrient limitation of phytoplankton[13], near-term ToE assessments suggest that variables related to nitrogen cycling, such as surface nitrate concentrations and NPP, only show detectable trends near the end of the twenty-first century, if at all[10,16]. This highlights the challenge of detecting and predicting change in the nitrogen cycle, which is underpinned by a suite of overlapping biological processes that are highly sensitive to environmental change and may show strong variability. For instance, key nitrogen cycle processes will be altered directly as warming changes biogeochemical rates (i.e., metabolism)[17,18] and indirectly as circulation alters substrate supply[19]. Identifying the primary drivers and responses of the nitrogen cycle would facilitate improved understanding of the evolution of its source-sink balance and provide greater confidence in long-term projections of key ecosystem flows, such as NPP.

Nitrogen isotopes offer a sensitive means to detect such shifts. The isotopic composition of nitrogen ($\delta^{15}N$, where $\delta^{15}N = ({}^{15}N{:}^{14}N_{sample}/{}^{15}N{:}^{14}N_{standard} - 1) \times 1000$) is altered uniquely by different processes, such that nitrogen isotopes can record changes in the sources (biological nitrogen fixation and land-to-sea fluxes, including river input and atmospheric deposition), sinks (denitrification in the sediments and water column) and internal transformations (e.g., phytoplankton assimilation) of nitrogen. In general, nitrogen assimilation by phytoplankton and nitrogen sinks increase $\delta^{15}N$ of nitrate ($\delta^{15}N_{NO3}$), whereas sources of nitrogen decrease $\delta^{15}N_{NO3}$[20–22]. However, increases associated with phytoplankton assimilation are also modulated by the extent to which nitrogen is limiting, with phytoplankton in nitrogen-limited systems showing much weaker preference for the light isotope[21,23,24]. This utilization effect results in peak $\delta^{15}N_{NO3}$ values at the boundary between nitrogen-replete and nitrogen-limited regimes, and the lateral transport of ${}^{15}N$-enriched nitrogen into the gyres[20,22,25]. Palaeoceanographers have long used nitrogen isotopes to probe past changes in nitrogen cycling and utilization[26–28], and yet how nitrogen isotopes fingerprint anthropogenic impacts over the twenty-first century is unknown.

Here we use a global ocean-biogeochemical model[29] with nitrogen isotopes (Supplementary Notes 1 and 2, and Supplementary Figs. 1–3) to investigate the anthropogenic perturbation of the marine nitrogen cycle in the twenty-first century. We quantify the relative roles of climate change under the high

emissions Representative Concentration Pathway 8.5 (RCP8.5) scenario[30] and anthropogenic increases in nitrogen deposition[8], assess the ToE of nitrogen isotopes and disentangle the direct (warming on biogeochemical rates) and indirect (substrate supply on biogeochemical rates) effects of climate change to reveal the primary drivers of change. We find that nitrogen isotopes fingerprint the dominant role of climate change, specifically how circulation changes can intensify nitrogen limitation of lower latitude ecosystems, leading to decreased NPP and nitrogen sinks, and subsequently alter the bioavailable nitrogen budget.

## Results

**Anthropogenic alteration.** To assess the impact of atmospheric nitrogen deposition and climate change on the marine nitrogen cycle and its isotopes, we performed four simulations using the Pelagic Interactions Scheme for Carbon and Ecosystem Studies version 2 (PISCES-v2) biogeochemical model[29] forced by output from the Institut Pierre-Simon Laplace Climate Model 5A (IPSL-CM5A). These simulations included a preindustrial control scenario (1801–2100), an anthropogenic scenario that combined historical (1851–2005) and future projections (2006–2100) of atmospheric nitrogen deposition and climate change (using the high emissions RCP8.5 scenario), as well as two additional simulations with each anthropogenic driver considered in isolation (see 'Methods'). The anthropogenic effects on nitrogen cycling were quantified by comparing mean conditions at the end of the twenty-first century (2081–2100) with mean conditions at the end of the preindustrial control simulation.

The combination of anthropogenic nitrogen deposition and climate change had strong effects on the modelled nitrogen cycle when compared to the preindustrial control. Nitrogen deposition[8] increased from 16.1 Tg N yr$^{-1}$ between 1801 and 1850 to peak at 41.9 Tg N yr$^{-1}$ by 2030, and declined thereafter to 40.0 Tg N yr$^{-1}$ by 2081–2100 (Fig. 1a and Supplementary Fig. 4). The Northern Hemisphere received the highest deposition rates, in particular the west Pacific in agreement with recent observations[7,31]. New nitrogen supply from biological nitrogen fixation declined from 78.8 to 73.9 Tg N yr$^{-1}$ (−6% of its preindustrial rate) by 2081–2100, consistent with other ESM simulations that considered increasing nitrogen deposition[3,32–36], and displayed a clear tropical to subtropical shift (Fig. 1b). Declines in NPP of between 5% and 60% developed across the lower latitudes by 2081–2100, which drove a global decline in nitrogen consumption by phytoplankton of 297 Tg N yr$^{-1}$ (−5%), consistent with other ESM projections[10,11,37,38] (Fig. 1c). Declines in nitrogen utilization by zooplankton grazing (Fig. 1d) were consistent with trophic amplification[39], whereas denitrification changes (in both the water column and sediments; Fig. 1e, f) depended on local changes in particulate organic matter export, a dependency consistent with field investigations[40] and data-constrained modelling[36,41]. Overall, the combination of nitrogen deposition and climate change increased bioavailable nitrogen sources by 19.2 Tg N yr$^{-1}$, whereas decreasing bioavailable nitrogen sinks by 8.3 Tg N yr$^{-1}$, resulting in net gains of 27.5 Tg N yr$^{-1}$ by 2081–2100 (Fig. 1g). Although these gains in the global marine nitrogen budget are greater than those reported previously[3], a common inter-model response to anthropogenic impacts appears to be a shift towards nitrogen accumulation.

An important point is that climate change dominated the alteration of the marine nitrogen budget. By 2081–2100, climate change had increased the bioavailable nitrogen budget by 23.7 Tg N yr$^{-1}$ in the absence of historical and future increases in nitrogen deposition. This increase is explained by an increase in nitrogen fixation (+7.0 Tg N yr$^{-1}$) and a decrease in sinks (denitrification (−13.6 Tg N yr$^{-1}$) and burial (−3.1 Tg N yr$^{-1}$); Supplementary

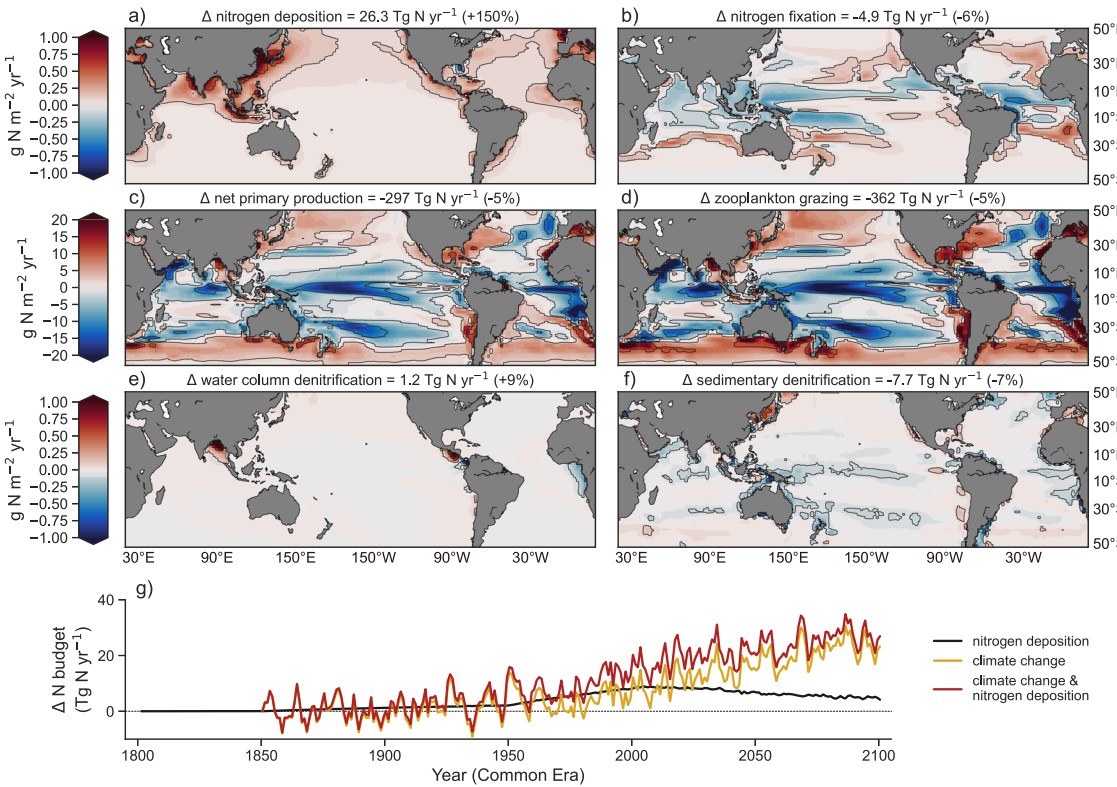

**Fig. 1 Anthropogenic perturbation (Δ) of the marine nitrogen cycle by 2081–2100.** Change is relative to the preindustrial control simulation. Top row shows major sources: **a** nitrogen deposition; **b** nitrogen fixation. Upper middle row shows internal cycling: **c** net primary production; **d** zooplankton grazing. Lower middle row shows major sinks: **e** water column denitrification; **f** sedimentary denitrification. Burial of nitrogen in sediments is not shown. Globally integrated changes are shown above each panel. Note the different scale for internal cycling. **g** Change in global nitrogen budget (sources minus sinks) due to anthropogenic drivers.

Fig. 5). In contrast, the anthropogenic increase in nitrogen deposition without climate change led to a small change in the budget of +4.9 Tg N yr⁻¹ (Fig. 1g) due to strong compensatory feedbacks, consistent with other ESMs[33–35], wherein newly deposited nitrogen either replaced nitrogen previously provided by nitrogen fixation (−12.0 Tg N yr⁻¹) or was rapidly removed by a local acceleration of denitrification (+6.4 Tg N yr⁻¹) and burial (+2.8 Tg N yr⁻¹; Supplementary Fig. 6). The individual effects of climate change (+23.7 Tg N yr⁻¹) and nitrogen deposition (+4.9 Tg N yr⁻¹), while not perfectly additive, combined to cause the net accumulation of nitrogen in the ocean (+27.5 Tg N yr⁻¹).

**Isotopic signals.** By 2081–2100, the combination of climate change and nitrogen deposition caused widespread, detectable declines in $\delta^{15}N$ of nitrate ($\delta^{15}N_{NO3}$; Fig. 2) and particulate organic matter ($\delta^{15}N_{POM}$; Fig. 3a) across the lower latitude ocean relative to the historical period (1986–2005), when most baseline isotopic measurements were taken[20] and against which twenty-first century trends might be assessed. Declines in $\delta^{15}N_{POM}$ developed in both the euphotic[42] and twilight zones (see 'Methods' and Supplementary Fig. 7), whereas declines in $\delta^{15}N_{NO3}$ were clearest in the twilight zone (Fig. 2a and Supplementary Fig. 8). Changes in $\delta^{15}N_{NO3}$ in the euphotic zone were highly variable, as small changes in nitrogen cycling amid low nitrate concentrations altered ¹⁵N:¹⁴N ratios substantially. In contrast, twilight zone $\delta^{15}N_{NO3}$ declined more uniformly by >0.2‰ across the tropical and subtropical Pacific and the Atlantic by 2081–2100 (Fig. 2a), with stronger signals in the gyres where nitrate concentrations are lowest (Supplementary Fig. 9). Unlike the euphotic zone, these declines in the twilight zone are within detection limits of observational methods,

as concentrations of nitrate exceed the 0.3 mmol m⁻³ threshold required for isotopic measurement[43].

Declines in nitrogen isotopes emerged more rapidly from background variability than other metrics related to the nitrogen cycle. By 1960–2000, trends in twilight zone $\delta^{15}N_{NO3}$ had emerged over much of the North Pacific, western equatorial Pacific, Arabian Sea and Atlantic Oceans (Fig. 2b). By year 2020, both $\delta^{15}N_{NO3}$ and $\delta^{15}N_{POM}$ had emerged across 50% of the lower latitude ocean, increasing to 77–82% by 2081–2100 (Fig. 2c). Isotopic trends developed earlier and were more widespread than other variables, including $NO_3$ concentrations and the N* tracer (N* = nitrate–phosphate · 16) in both the euphotic and twilight zones, as well as vertically integrated NPP and nitrogen fixation (Fig. 2c).

Although both contributed to the isotopic declines, the influence of climate change exceeded nitrogen deposition during the twenty-first century. The impact of nitrogen deposition on $\delta^{15}N_{NO3}$ began in 1950 and showed little additional effect after 2050 in our model (Fig. 2d–f). In contrast, climate change impacts were only noticeable after 1975 and emerged across 60% of the lower latitude ocean in the twenty-first century (Fig. 2g–i). By 1986–2005, nitrogen deposition was responsible for emergent $\delta^{15}N_{NO3}$ trends across 29–41% of the lower latitude twilight zone (Fig. 2f), compared with only 3–10% due to climate change (Fig. 2i). In the northwest Pacific, a strong decline in $\delta^{15}N_{POM}$ recorded in corals has been attributed to the rise in nitrogen deposition in recent decades[31] and our simulations support this attribution. However, in the four decades from 2020–2060 (dotted areas), the pattern was reversed, with 31.9% of the ocean showing emergent trends due to climate change compared to only 14.2% due to nitrogen deposition. The dominance of climate

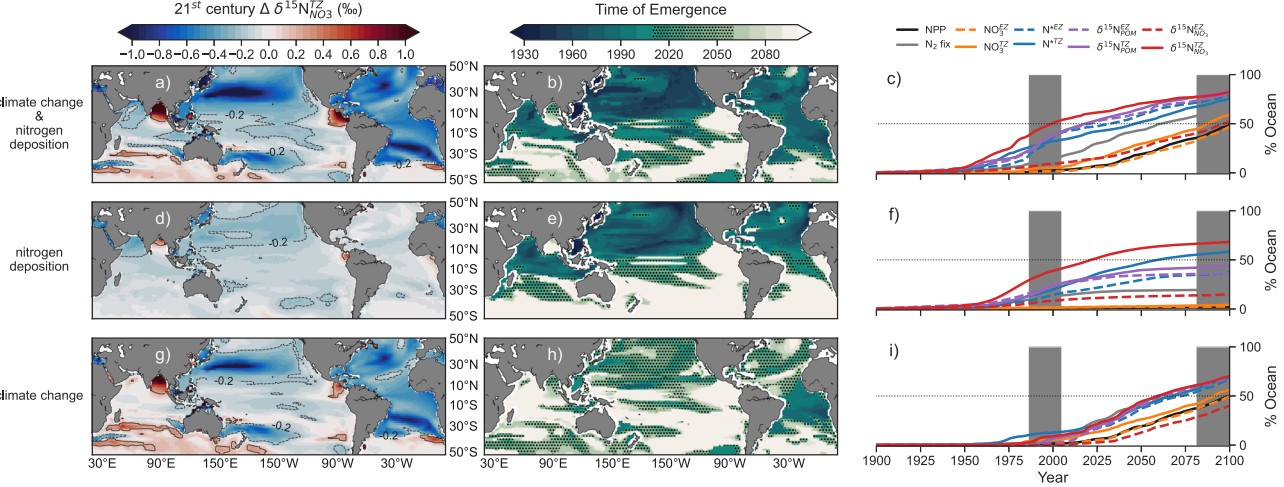

**Fig. 2 Anthropogenic trends of nitrogen isotopes in the twilight zone and their Time of Emergence. a** End of twenty-first century (2081–2100) minus turn of the century (1986–2005) twilight zone $\delta^{15}N_{NO3}$ due to both anthropogenic drivers. **b** Time of Emergence of twilight zone $\delta^{15}N_{NO3}$. **c** Percentage of the low-latitude ocean (45°S–45°N) with emergent trends in biogeochemical variables related to the marine nitrogen cycle. They include vertically integrated net primary production (NPP) and nitrogen fixation ($N_2$ fixation), nitrate ($NO_3$) concentrations in both the euphotic zone (EZ) and twilight zone (TZ), $\delta^{15}N_{NO3}$ and $\delta^{15}N_{POM}$. **d–f** Same as **a–c** but for experiment with nitrogen deposition only. **g–i**, Same as **a–c** but for experiment with climate change only.

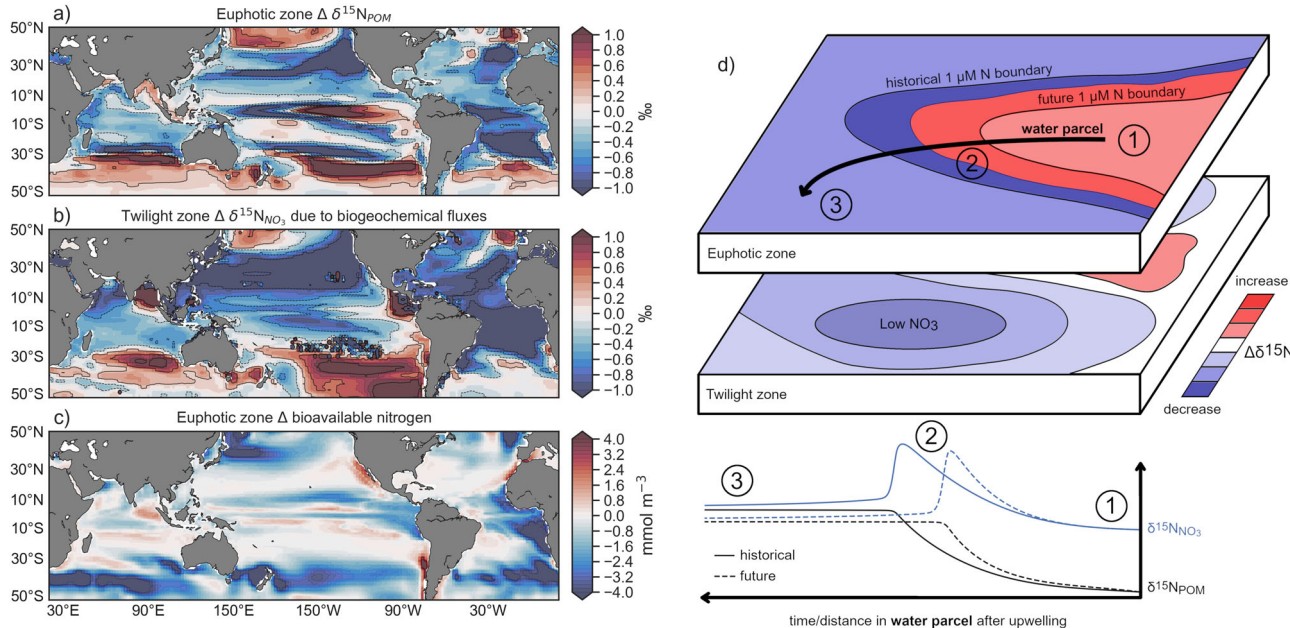

**Fig. 3 Biogeochemical control on the nitrogen cycle perturbation. a** Climate change effect on euphotic zone $\delta^{15}N_{POM}$. **b** Climate change effects on twilight zone $\delta^{15}N_{NO3}$ due to biogeochemical processes only. **c** Change in euphotic zone bioavailable nitrogen (nitrate plus ammonium). **d** Schematic describing a major mechanism of $\delta^{15}N$ depletion in our experiments. Less upwelled bioavailable nitrogen (N) within a water parcel travelling from an upwelling zone (1) to the subtropical gyres (3) leads to lower $\delta^{15}N_{POM}$ outside of the productive zone. Greater nitrogen limitation of phytoplankton generates less enrichment of $^{15}N$ in unused nitrogen (1), leading to a lower peak in $\delta^{15}N_{NO3}$ and $\delta^{15}N_{POM}$ at the boundary between nitrogen-replete and nitrogen-limited regimes (2). A sharp depletion in $\delta^{15}N_{NO3}$ is recorded in the area where the nitrogen-replete to nitrogen-limited transition was previously located under historical conditions. This depleted signal is carried laterally into the nitrogen-limited gyres (3) and delivered to the twilight zone via remineralization of low $\delta^{15}N_{POM}$. Future $\delta^{15}N_{POM}$ may be higher in regions near to upwelling (equatorial Pacific and Benguela upwelling in **a**), because nitrogen limitation decreases fractionation by phytoplankton, meaning that more $^{15}N$ is assimilated into organic matter and removed from the euphotic zone. The schematic in **d** was made using GIMP version 2.10.6 (https://www.gimp.org/).

change continued to 2081–2100 with an additional emergence of 10–20% (Fig. 2i), compared to only 2–4% due to nitrogen deposition (Fig. 2f). Climate change was particularly important for emergences in the Southern Hemisphere (Fig. 2g) where rates and changes in nitrogen deposition were minimal (Fig. 1a).

**Linking nitrogen cycling and isotopic signals**. Although low $\delta^{15}N$ from nitrogen deposition was important for twentieth century declines in $\delta^{15}N_{NO3}$, consistent with previous modelling[35], climate-driven declines in $\delta^{15}N_{NO3}$ during the twenty-first century potentially involved changes to numerous

nitrogen cycle processes, the individual effects of which are difficult to isolate. Potential contributors include nitrogen limitation of phytoplankton, which limits the rate and strength of their fractionation, an increase in nitrogen fixation, an increase in zooplankton recycling, a decrease in denitrification or a physical redistribution of the dissolved nitrogen compounds by a changing circulation (or any combination of these processes). Here we demonstrate that changes ultimately linked to increasing nitrogen limitation were responsible for the broad, simulated $\delta^{15}N_{NO3}$ declines that emerged in the twilight zones of the low-latitude oceans. Namely, a decrease in phytoplankton production and fractionation potentially supplemented by a tropical–subtropical shift of nitrogen fixation.

First, the presence of $^{15}$N-depleted organic matter sinking from the overlying euphotic zone is a first-order driver of the $\delta^{15}N_{NO3}$ declines in the twilight zone. Widespread declines in euphotic zone $\delta^{15}N_{POM}$ (Fig. 3a) thus delivered less $^{15}$N to twilight zones across large parts of the lower latitude ocean following the remineralization of sinking organic material. This driver was further supported by an offline analysis of purely biogeochemical $^{15}NO_3$ fluxes in our global model. The only internal biogeochemical source of nitrate is from nitrification of the ammonium that forms following remineralization, whereas the main sinks are primary production and denitrification. By isolating the fluxes of these individual sources and sinks within each grid cell, we disentangled local biogeochemical effects from circulation effects (see 'Methods'). Biogeochemical fluxes tended to deplete $\delta^{15}N_{NO3}$ across the tropics and subtropics, and the ensuing changes to $\delta^{15}N_{NO3}$ were greater than observed in the full model (Fig. 3b). This indicates that physical sources and sinks (i.e., ocean transports) partially compensated for the biogeochemical effects. Smoothing of the strong gradients set by biogeochemical fluxes is consistent with the role of ocean physics for upward mixing of deep nitrate with relatively constant $\delta^{15}N_{NO3}$ values. The only exceptions to this picture were denitrification zones in the North Indian and East Pacific Oceans, where denitrification rates increased and raised $\delta^{15}N_{NO3}$, and in South Indian and South Pacific downstream of Southern Ocean mode waters[44,45], where upstream increases in NPP (Fig. 1c) raised $\delta^{15}N_{NO3}$. The upstream isotopic enrichment of mode waters exceeded the declines caused by nutrient limitation in the Indian Ocean and partially compensated for the declines caused by nutrient limitation in the Pacific. South Atlantic twilight zones were not affected, as its mode waters form at higher latitudes in the southeast Pacific and on deeper, denser isopycnals[45], which outcrop poleward of Subantarctic increases in NPP. Despite these regional exceptions, the broad biogeochemical tendency was to lower twilight zone $\delta^{15}N_{NO3}$ and this was linked to depleted $\delta^{15}N_{POM}$ sinking out of subtropical euphotic zones.

Second, the decline in nitrogen assimilation by phytoplankton (by 297 Tg N yr$^{-1}$ globally; Fig. 1c) far exceeded parallel changes to the major sources and sinks, and made an important contribution to the simulated isotopic declines. Increasing nitrogen limitation of phytoplankton reduced nitrogen assimilation across the lower latitude oceans during the twenty-first century, with bioavailable nitrogen declining by 0.5–4 mmol m$^{-3}$ (5–40%) in the tropical Pacific and Atlantic euphotic zones (by 2081–2100 relative to 1986–2005; Fig. 3c). These trends are consistent with the common projections of declining upper-ocean nitrate inventories across ESM studies that are largest in the low-latitude upwelling systems[10,11]. This is important, because changing nitrogen availability not only affects the success of nitrogen fixers, but also affects the rate of nitrogen uptake by phytoplankton and the degree of nitrogen isotope fractionation, which combine to control the horizontal $\delta^{15}N_{NO3}$ gradients across the low-latitude ocean[1,20,22–24]. Increasing nitrogen

limitation under climate change therefore led to both a stimulation of nitrogen fixation and a weaker isotopic enrichment associated with phytoplankton assimilation. Weaker fractionation by phytoplankton (i.e., a weaker preference for $^{14}$N) meant that more $^{15}$N was used to create organic matter in increasingly nitrogen-limited upwelling regions and was subsequently lost via sinking (step 1 in Fig. 3d). Consequently, euphotic zone $\delta^{15}N_{NO3}$ and $\delta^{15}N_{POM}$ values declined at the boundary between the nitrogen-replete and nitrogen-deplete regimes (step 2 in Fig. 3d). As subtropical gyres receive most of their nutrients from lateral transport[25], $^{15}$N-depleted nitrogen was then swept into the subtropical gyres where local nitrogen recycling and organic matter formation proceeded with lighter isotopic signatures (i.e., relatively more $^{14}$N; step 3 in Fig. 3d), and, combined with local increases in biological nitrogen fixation, delivered lower $\delta^{15}N_{POM}$ values to the subtropical twilight zone.

The removal of $^{15}$N by nitrogen-limited phytoplankton in the tropics followed by the transfer of $^{15}$N-depleted water to the gyres is also supported by idealized modelling. We constructed a zero-dimensional (0D) model that follows nitrogen uptake and fractionation by phytoplankton in a water parcel (see 'Methods,' Supplementary Note 3, Supplementary Table 1 and Supplementary Fig. 10). In this simple model, an inorganic nitrogen pool representing initial upwelled nitrogen is steadily assimilated by phytoplankton to create organic matter and this organic matter is either remineralized back to inorganic nitrogen or removed permanently via export. For every 10% decline in initial upwelled nitrogen supplied to the water parcel (corresponding to a ~10% loss in integrated NPP during the water parcel's lifetime), there is a 0.19 ± 0.03‰ decline in the final $\delta^{15}N_{POM}$ produced within the gyre once nitrogen is depleted to limiting concentrations. The uncertainty of ±0.03‰ is associated with temperature changes of ±4 °C, suggesting that warming also plays a role by modulating growth rates and recycling (Supplementary Fig. 11). Extrapolating this 'rule of thumb' to the global model, we expect $\delta^{15}N_{POM}$ declines of 0.1–0.8‰ for bioavailable nitrogen declines of 5–40%, which agrees broadly with the results of the full model by 2081–2100 (Fig. 2c). Furthermore, the local enrichment of $\delta^{15}N_{POM}$ in the upwelling region in both our 0D model (compare solid and dashed lines in Fig. 3d) and in the tropical Pacific in our global model (Fig. 3a) clearly signifies the existence of this mechanism, where more $^{15}$N was removed from upwelling zones due to nitrogen limitation of phytoplankton (it is noteworthy that this feature was absent in the Atlantic due to local declines in denitrification; Fig. 1g). Ultimately, the biogeochemical consequences of increasing nitrogen limitation appeared to be the primary cause of the widespread isotopic declines.

**Warming vs. circulation changes**. We examined the direct (i.e., warming on biogeochemical rates) and indirect effects (i.e., circulation altering nutrient supply) of climate change in two additional experiments. First, warming was imposed on biogeochemical processes under otherwise preindustrial conditions to mimic its effect on rates. Second, preindustrial temperatures were imposed, while climate change altered the circulation to mimic its effect on substrate availability (see 'Methods'). Direct effects of warming on biogeochemical processes showed a limited ability to reproduce the full suite of climate-driven trends, with the exception of the poleward shift in nitrogen fixers (Fig. 4). In contrast, changes to ocean circulation (e.g., stratification) in the indirect effect simulation replicated changes in euphotic zone nitrate (Spearman's rank correlation; $r_s = 0.99$), twilight zone $\delta^{15}N_{NO3}$ ($r_s = 0.94$), twilight zone $\delta^{15}N_{POM}$ ($r_s = 0.95$), NPP ($r_s = 0.69$), zooplankton grazing ($r_s = 0.69$) and water column denitrification ($r_s = 0.88$) and sedimentary denitrification ($r_s =$

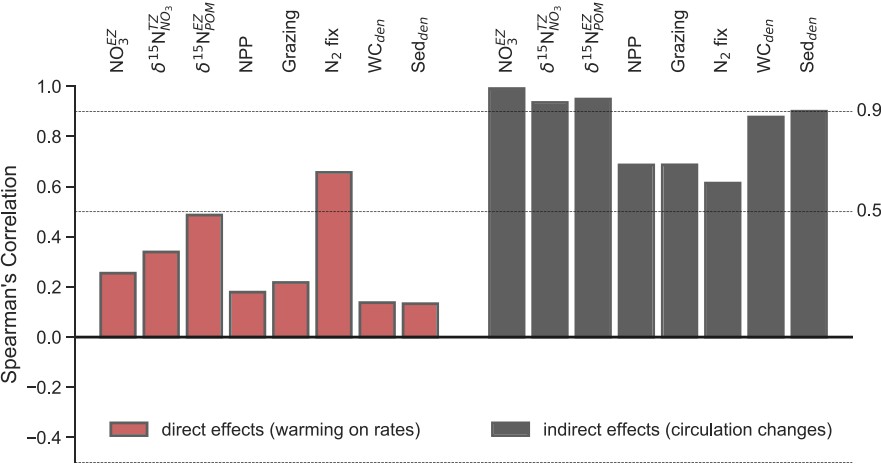

**Fig. 4 Direct and indirect effects of climate change on the nitrogen cycle.** How well the direct effects (warming on biogeochemical rates) and indirect effects (circulation changes, including stratification) of climate change reproduce changes in the nitrogen cycle and its isotopes in the full model. Agreement is measured as a cell-to-cell comparison using Spearman's rank correlation between the climate change experiment and these experiments, comparing 2D fields of euphotic zone nitrate ($NO_3^{EZ}$), twilight zone $\delta^{15}N_{NO3}$ ($\delta^{15}N_{NO_3}^{\varnothing}$), euphotic zone $\delta^{15}N_{POM}$ ($\delta^{15}N_{POM}^{EZ}$), depth integrated net primary production (NPP), zooplankton grazing (Grazing), biological nitrogen fixation ($N_2$ fix), water column denitrification ($WC_{den}$) and sedimentary denitrification ($Sed_{den}$) at 2081–2100.

0.90) seen in the full model (Fig. 4). Moderate agreement for nitrogen fixation in both the direct effect ($r_s = 0.65$) and indirect effect simulations ($r_s = 0.61$) suggested that both warming and circulation changes were equally important in determining its full response, but that nitrogen fixation by itself was not sufficient to force the isotopic trends associated with climate change. The effect of climate change on biogeochemical processes via altering substrate availability was therefore a major cause of twenty-first century increases in nitrogen limitation, resultant NPP declines, shifts in the sources and sinks of nitrogen, and the accompanying trends in nitrogen isotopes that fingerprint the response of the nitrogen cycle.

## Discussion

Our experiments demonstrate the dominant role of climate change in the twenty-first century alteration of the marine nitrogen cycle, despite a 2.5-fold increase in atmospheric nitrogen deposition since the preindustrial period. Central to the climate-driven perturbation was increasing nitrogen limitation of phytoplankton in the low latitudes due to changes in ocean circulation, likely due to increasing stratification, which is ongoing[12,46] and expected to intensify this century[10,11,19,37]. Importantly, these changes are clearly fingerprinted by detectable declines in the isotopic composition of nitrate and particulate organic matter within twilight zones across much of the lower latitude ocean (Fig. 3d). Although repeat $\delta^{15}N_{NO3}$ measurements along hydrographic lines are currently lacking, future occupations may detect the effects of climate change. This is because the currently available measurements of $\delta^{15}N_{NO3}$ in the twilight zones of the Pacific ($N = 1481$) and Atlantic ($N = 890$) have been made near peak rates of nitrogen deposition (median age of data = 2008)[20]. As future deposition is expected to plateau[4,7], any future trends should be dominated by climate-driven signals.

Our simulations agree with field studies[40] and data-constrained modelling[36,41] to highlight how the expected declines in low-latitude NPP[10] associated with increasing nitrogen limitation may reduce nitrogen sinks as less organic matter sinks out of euphotic zones. Although this may contradict expectations associated with an expansion of oxygen minimum zones[10,47–49], mesopelagic nitrogen loss processes are highly sensitive to organic matter flux[36,40,41]. A primary dependency on organic matter flux is reflected in our experiments, where nitrogen sinks declined despite a slight increase

in hypoxic ($O_2 < 80\ mmol\ m^{-3}$) water volume of 0.6% by 2081–2100, relative to preindustrial conditions. If nitrogen sinks do decline as less organic matter sinks out of the euphotic zone, then our results suggest that the bioavailable nitrogen reservoir may accumulate (Fig. 1g). Although the response of hypoxic zones, and hence nitrogen sinks, to climate change is subject to considerable uncertainty[50], the shift towards accumulation is consistent with previous modelling[3] and may be reinforced by including currently unrepresented processes, such as the stimulation of nitrogen fixation by increasing $pCO_2$[46] and anthropogenic iron deposition[51]. Net accumulation of bioavailable nitrogen in the twenty-first century, if realized, may mitigate or even counteract the multi-centennial declines in NPP that are projected as a result of nutrient trapping in the deep ocean[13] and potentially represents an important self-stabilizing mechanism for ocean productivity on multi-centennial timescales. Importantly, our results suggest that this response will be fingerprinted by $\delta^{15}N_{NO3}$ declines in the subtropical ocean twilight zone.

The ocean twilight zone is emerging as an important depth stratum to detect trends[52,53]. Signals may be larger in the euphotic zone, but detection is severely challenged by high-frequency variability[54,55]. In contrast, the twilight zone collects euphotic zone signals through sinking and remineralization of organic matter and acts as a low-pass filter to record large-scale, multi-annual changes in upper-ocean-biogeochemical processes. The relative stability of the twilight zone is demonstrated by the low variability of modern-day $\delta^{15}N$ recorded in sub-euphotic foraminifera[56]. As foraminifera are now routinely used to reconstruct $\delta^{15}N$ for studies of the past oceanic nitrogen cycle[21,26,28], our results suggest that sedimentary archives of sub-euphotic species may provide an opportunity to investigate past variations in nitrogen availability across the low latitudes. If we consider the coming decades, the strong sensitivity of nitrogen isotopes to large-scale changes in nitrogen cycling mean that repeat hydrographic surveys and multi-decadal time series can complement remote sensing and bioArgo floats to constrain the impacts of climate change and natural variability, providing an important means to assess model projections of NPP and an enigmatic marine nitrogen cycle.

## Methods

**Modelling approach.** We used the PISCES-v2 biogeochemical model, attached to the Nucleus for European Modelling of the Ocean version 4.0 (NEMO-v4) general ocean circulation model[29]. PISCES-v2 includes five nutrients pools (nitrate, ammonium,

phosphate, silicic acid and dissolved iron), dissolved oxygen, the full carbon system and accounts for two phytoplankton (nanophytoplankton and diatoms) and two zoo-plankton types (microzooplankton and mesozooplankton). Bioavailable nitrogen in our simulations is considered to be the combination of nitrate and ammonium. Its nitrogen cycle includes nitrogen fixation, nitrification, burial, denitrification in both the water column and sediments, and coupled nitrification–denitrification. Nitrogen isotopes were integrated within PISCES-v2 for the purposes of this study, using nine new tracers (Supplementary Note 1). Horizontal model resolution varied between ~0.5° at the equator and poles, and 2° in the subtropics, whereas vertical resolution varied between 10 and 500 m thickness over 31 levels.

We conducted simulations under both preindustrial control and climate change scenarios. The preindustrial control scenario from 1801 to 2100 maintained preindustrial greenhouse gas concentrations and only included internal modes of variability. The climate change simulation from 1851 to 2100 included natural variability, prescribed changes in land use, as well as historical changes in concentrations of greenhouse gases and aerosols until 2005, after which future concentrations associated with RCP8.5 were imposed[30]. The biogeochemical model (PISCES-v2) was run offline from the physical model (NEMO-v4) using monthly transports and other physical conditions generated by the low resolution version of the IPSL-CM5A ESM[57].

Experiments were initialized from biogeochemical fields created from an extensive spin-up of 5000 years under repeat physical forcing, followed by a 300-year simulation under the preindustrial control scenario. The preindustrial control simulation used in analysis was therefore the final 300 years of a 5600-year spin-up involving two repeat simulations of the preindustrial control scenario. We utilized a global compilation of $\delta^{15}N_{NO3}$[20] supplemented with recent data to assess the isotopic routines in the model and conducted a thorough model-data skill assessment at replicating observed patterns in space (Supplementary Note 2 and Supplementary Figs. 1–3).

**Anthropogenic nitrogen deposition**. The effect of increasing aeolian deposition of nitrogen was assessed in our simulations. Preindustrial nitrogen deposition was prescribed as the preindustrial estimate at 1850, whereas the historical to future deposition was created by linear interpolation between preindustrial (1850) and modern/future fields (2000, 2030, 2050 and 2100). These fields were provided by Hauglustaine et al.[8]. However, the rapid rise between 1950 and 2000 was maintained, such that 60% of the increase between the preindustrial and modern fields occurred after 1950 (Supplementary Fig. 4).

The historical rise in anthropogenic nitrogen deposition was assessed by including it in additional simulations under both preindustrial control and climate change scenarios. Four initial experiments were therefore conducted: preindustrial control; preindustrial control plus anthropogenic nitrogen deposition; climate change; and climate change plus anthropogenic nitrogen deposition.

**Global model experiments**. We undertook four initial simulations to quantify the impacts of anthropogenic climate change and nitrogen deposition: a preindustrial control simulation from 1801 to 2100; a full anthropogenic scenario from 1851 to 2100; a climate change-only scenario without the increase in anthropogenic nitrogen deposition from 1851 to 2100; and a nitrogen deposition scenario without anthropogenic climate change from 1851 to 2100. Anthropogenic effects to nitrogen cycling were quantified by comparing mean conditions over the final 20 years of the twenty-first century (2081–2100) with mean conditions over the final 20 years of the pre-industrial control simulation, whereas effects on nitrogen isotopes were quantified by comparing mean conditions over the final 20 years of the twenty-first century (2081–2100) with mean conditions over the historical period (1986–2005) from the same simulation.

To understand the direct and indirect effects of climate change, we undertook two additional idealized simulations. First, we imposed temperature changes on biogeochemical rates, while maintaining ocean circulation associated with the preindustrial control scenario, to assess the direct effects of warming on biogeochemical processes. Second, we imposed the preindustrial control temperature field on biogeochemical processes, while altering the circulation in line with the climate change scenario, to assess the indirect effects of climate change (i.e., how changing circulation alters substrate supply to biogeochemical reactions). Each experiment was run from 1851 to 2100 and without the anthropogenic increase in atmospheric nitrogen deposition, parallel with the full climate change simulation.

Agreement between the climate change simulation without anthropogenic nitrogen deposition was quantified using a pixel-by-pixel correlation analysis using Spearman's rank correlation based on the non-parametric nature of the two-dimensional fields used for comparison. Fields were euphotic zone nitrate, twilight zone $\delta^{15}N_{NO3}$, euphotic zone $\delta^{15}N_{POM}$, and vertically integrated NPP, zooplankton grazing, nitrogen fixation, water column denitrification and sedimentary denitrification.

**Depth zones**. We assessed changes in biogeochemical variables related to nitrogen cycling in two depth zones defined by light. The euphotic zone was defined by depths between the surface and 0.1% of incident irradiance as recommended by Buessler et al.[42]. The twilight zone was also defined using light, as advocated by Kaartvedt et al.[58]. Depths between 0.1% and 0.0001% of incident irradiance defined the twilight zone. These definitions typically returned euphotic zone thicknesses of 137 ± 23 m (mean ± SD), and twilight zone thicknesses of 233 ± 37 m. The

boundary between these depth zones were deepest in oligotrophic tropical and subtropical waters, and were shallowest in equatorial and temperate waters (Supplementary Fig. 7).

**Time of emergence**. ToE calculations determined when anthropogenic, anomalous trends emerged from the noise of background variability. ToE was calculated at each grid cell within both the euphotic and twilight zones (depth-averaged) and using annually averaged fields of ocean tracers. We therefore ignored temporal trends and variability at seasonal and sub-seasonal scales. Raw time series were first detrended and normalized using the linear slope and mean of the preindustrial control experiment, such that the preindustrial control time series varied about zero, while anomalous trends in experiments with climate change and/or nitrogen deposition deviated from zero. These detrended and normalized time series were smoothed using a boxcar (flat) moving average with a window of 11 years to filter decadal variability (Supplementary Fig. 12). Differences with the preindustrial control experiment were then computed.

To determine whether the differences with the preindustrial control experiment were anomalous, we calculated a measure of noise from the raw, inter-annual time series of the preindustrial control experiment (1801–2100). A signal emerged from the noise if it exceeded 2 SDs, a threshold that represents with 95% confidence that a value was anomalous and is therefore a conservative envelope to distinguish normality from anomaly[16].

Furthermore, we required that anomalous values must consistently exceed the noise of the preindustrial control experiment until the end of the simulation (2100) to be registered as having emerged. Temporary emergences were therefore rejected, making our ToE estimates more conservative. A graphical representation of this process is shown in Supplementary Fig. 12.

**Isolating biogeochemical $^{15}NO_3$ fluxes**. We analysed the biogeochemical fluxes of $^{15}NO_3$ and $NO_3$ into and out of each model grid cell within the twilight zone, to determine whether the trends in $\delta^{15}N_{NO3}$ were related to biogeochemical or physical changes. Fluxes of $^{15}NO_3$ and $NO_3$ included a net source from nitrification ($NO_3^{nitr}$) and net sinks due to new production ($NO_3^{new}$) and denitrification ($NO_3^{den}$). Although nitrification did not directly alter the $^{15}N:^{14}N$ ratio in our simulations, the release of $^{15}NO_3$ and $NO_3$ by nitrification conveyed an isotopic signature determined by prior fractionation processes that produce ammonium ($NH_4$). These processes include remineralization of particulate and dissolved organic matter, excretion by zooplankton and nitrogen fixation. The isotopic signatures of these processes were thus included implicitly in $NO_3^{nitr}$. For each grid cell, we calculated the biogeochemical tendency to alter $\delta^{15}N_{NO3}$ based on the ratio of inputs minus outputs:

$$\Delta\delta^{15}N_{NO3} = \left( \frac{^{15}NO_3^{nitr} - ^{15}NO_3^{new} - ^{15}NO_3^{den}}{^{14}NO_3^{nitr} - ^{14}NO_3^{new} - ^{14}NO_3^{den}} - 1 \right) \cdot 1000 \tag{1}$$

This calculation excluded any upstream biological changes and circulation changes that might have altered $\delta^{15}N_{NO3}$.

**0D water parcel model**. We simulated the nitrogen isotope dynamics in a recently upwelled water parcel during transit to the subtropics by building a 0D model. The model simulates state variables of dissolved inorganic nitrogen (DIN), particulate organic nitrogen (PON) and exported particulate nitrogen (ExpN), as well as their heavy isotopes (DI$^{15}$N, PO$^{15}$N and Exp$^{15}$N) in units of mmol N m$^{-3}$ over 100 days given initial conditions and constants listed in Supplementary Table 1.

$$\frac{\Delta DIN}{\Delta t} = -N_{uptake} + N_{recycled} \tag{2}$$

$$\frac{\Delta PON}{\Delta t} = N_{uptake} - N_{recycled} - N_{exported} \tag{3}$$

$$\frac{\Delta ExpN}{\Delta t} = N_{exported} \tag{4}$$

$$\frac{\Delta DI1^{15}N}{\Delta t} = -^{15}N_{uptake} + ^{15}N_{recycled} \tag{5}$$

$$\frac{\Delta PO^{15}N}{\Delta t} = ^{15}N_{uptake} - ^{15}N_{recycled} - ^{15}N_{exported} \tag{6}$$

$$\frac{\Delta Exp^{15}N}{\Delta t} = ^{15}N_{exported} \tag{7}$$

First, the model calculates maximum potential growth rate of phytoplankton ($\mu_{max}$) in units of day$^{-1}$ (Eq. 8) using temperature and then finds nitrogen uptake ($N_{uptake}$, Eq. 10) using PON and limitation terms for nitrogen ($N_{lim}$, Eq. 9), light ($L_{lim}$, Supplementary Table 1) and iron ($Fe_{lim}$, Supplementary Table 1).

$$\mu_{max} = 0.6 \, day^{-1} \cdot e^{T \cdot T_{growth}} \tag{8}$$

$$N_{lim} = \frac{DIN}{DIN + K_{DIN}} \tag{9}$$

$$N_{uptake} = \mu_{max} \cdot L_{lim} \cdot \min(Fe_{lim}, N_{lim}) \cdot PON \qquad (10)$$

At a constant temperature of 18 °C, $\mu_{max}$ is equal to ~1.9 day$^{-1}$. Limitation terms for light and iron are set as constant and are used to prevent unrealistically high nitrogen uptake when nitrogen is high, such as occurs immediately following upwelling in the high-nutrient low-chlorophyll regions of the tropics. Fractionation by phytoplankton is calculated assuming an open system[21], in this case where nitrogen can be lost through export of organic matter. To calculate the fractionation associated with uptake ($^{15}N_{uptake}$, Eq. 11), we multiply the total nitrogen uptake ($N_{uptake}$, Eq. 10) by the heavy to light isotope ratio ($r^{15}_{DIN}$, Eq. 12) and the fractionation factor ($\varepsilon_{phy}$, Supplementary Table 1), which is converted from units of per mil (‰) to a fraction relative to one. This fractionation factor ($\varepsilon_{phy}$) is constant at 5‰ but is decreased towards 0‰ by the nitrogen limitation term ($N_{lim}$, Eq. 9), such that when nitrogen is limiting to growth, the fractionation during uptake decreases (last term on the right-hand side approaches 1).

$$^{15}N_{uptake} = N_{uptake} \cdot r^{15}_{DIN} \cdot \left(1 - \frac{N_{lim} \cdot \varepsilon_{phy}}{1000}\right) \qquad (11)$$

$$r^{15}_{DIN} = \frac{DI^{15}N}{DIN} \qquad (12)$$

At each timestep, a fraction of the PON pool becomes detritus (Eq. 15) and this detritus is instantaneously recycled back to DIN or exported to ExpN and removed from the water parcel. The amount of detritus produced per timestep is calculated as the sum of linear respiration (Eq. 13) and quadratic mortality (Eq. 14) terms, where $P_{resp}$ (units of day$^{-1}$), $K_{resp}$ (units of mmol N m$^{-3}$) and $P_{mort}$ (units of (mmol N m$^{-3}$)$^{-1}$ day$^{-1}$) are constants (Supplementary Table 1).

$$Respiration = P_{resp} \cdot PON \cdot \frac{PON}{PON + K_{resp}} \qquad (13)$$

$$Mortality = P_{mort} \cdot PON^2 \qquad (14)$$

$$Detritus = Respiration + Mortality \qquad (15)$$

Once we know the fraction of PON that becomes detritus at any given timestep, we must solve for the fraction of that detritus that becomes DIN through recycling (Eq. 17), and that which becomes ExpN through export (Eq. 18). The fraction of detritus that is recycled back into DIN is temperature dependent (Eq. 16), with higher temperatures increasing rates of recycling above a minimum fraction set by $f_{recmin}$ (Supplementary Table 1). The relationship with temperature is exponential, similar to phytoplankton maximum growth ($\mu_{max}$), but the degree of increase associated with warming is scaled down by a constant factor equal to $T_{rec}$ (Supplementary Table 1). The fraction that is exported to ExpN is the remainder (Eq. 18).

$$f_{recycled} = f_{recmin} + T_{rec} \cdot e^{T \cdot T_{growth}} \qquad (16)$$

$$N_{recycled} = Detritus \cdot f_{recycled} \qquad (17)$$

$$N_{exported} = Detritus \cdot (1 - f_{recycled}) \qquad (18)$$

The major fluxes of $N_{uptake}$, $N_{recycled}$ and $N_{exported}$ are now solved for. All that remains is to calculate the isotopic signatures of the recycling (Eq. 19) and export (Eq. 20) fluxes. These, similar to $^{15}N_{uptake}$ (Eq. 11), are solved by multiplying against a standard ratio of heavy to light isotope ($r^{15}_{PON}$, Eq. 21).

$$^{15}N_{recycled} = N_{recycled} \cdot r^{15}_{PON} \qquad (19)$$

$$^{15}N_{exported} = N_{exported} \cdot r^{15}_{PON} \qquad (20)$$

$$r^{15}_{PON} = \frac{PO^{15}N}{PON} \qquad (21)$$

Finally, we calculate the $\delta^{15}N$ values of the major pools in the model (DIN, PON and ExpN) as output (Eqs. 22–24). We assume in this model that the major pools of DIN, PON and ExpN represent the total amount of the light isotope ($^{14}N$), whereas the DI$^{15}N$, PO$^{15}N$ and Exp$^{15}N$ pools represent the relative enrichment in $^{15}N$ compared to a standard ratio. For simplicity, we make the standard ratio equal to 1. Therefore, taking the ratio of the DI$^{15}N$ to DIN pools and subtracting one returns the isotopic signature. Multiplying this by 1000 converts this signature to per mil units (‰).

$$\delta^{15}N_{DIN} = \left(\frac{DI^{15}N}{DIN} - 1\right) \cdot 1000 \qquad (22)$$

$$^{15}N_{PON} = \left(\frac{PO^{15}N}{PON} - 1\right) \cdot 1000 \qquad (23)$$

$$^{15}N_{ExpN} = \left(\frac{Exp^{15}N}{ExpN} - 1\right) \cdot 1000 \qquad (24)$$

## Data availability

The data generated in this study have been deposited in the Zenodo database and are freely available at https://doi.org/10.5281/zenodo.5541332.

## Code availability

The version of NEMO-PISCES used in this study can be downloaded with the following command: svn co https://forge.ipsl.jussieu.fr/nemo/svn/NEMO/releases/release-4.0 NEMO4.0. Additional development to include the isotopes can be merged into this version by contacting the lead author. Analyses and code to produce graphics were conducted using both Ferret and Python, and are available on GitHub at https://github.com/pearseb/PISCESiso_Ncycle_analysis and are citable https://doi.org/10.5281/zenodo.5543510.

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

## Acknowledgements

P.J.B., A.T. and C.M. were supported by the ARISE project (NE/P006035/1), part of the Changing Arctic Ocean programme, jointly funded by the UKRI Natural Environmental Research Council (NERC) and the German Federal Ministry of Education and Research (BMBF). O.A. acknowledges support from the LEFE PISCO project. L.B. acknowledges support from the MERTOX (ANR-17-CE34-0010) and CIGOEF (ANR-17-CE32-0008) projects. Simulations were undertaken on Barkla, part of the High-Performance Computing facilities at the University of Liverpool. We acknowledge use of the Ferret program (http://ferret.pmel.noaa.gov/Ferret/), climate data operators (https://code.mpimet.mpg.de/projects/cdo/), NetCDF Operators (http://nco.sourceforge.net/), Python (www.python.org) and GIMP (https://www.gimp.org/) for the analysis and graphics in this study.

## Author contributions

The study was designed by P.J.B. and A.T. P.J.B., O.A. and L.B. developed the model. P.J.B. conducted all experiments and analysis. P.J.B., O.A., L.B., C.M. and A.T. interpreted the results. P.J.B. and A.T. led the writing of the manuscript. O.A., L.B. and C.M. edited the manuscript.

## Competing interests

The authors declare no competing interests.
