## [Peer Review File · Nature Communications]

Impact of intensifying nitrogen limitation on ocean net primary production is fingerprinted by nitrogen isotopesReviewers' Comments:

Reviewer #1:

Remarks to the Author:

In this study the authors want to assess the effect and relative contribution of atmospheric N deposition and climate change on the oceanic nitrogen cycle and on the resulting $\delta^{15}\text{N}$ isotopic signature over the 21st century.

The tool used is a global biogeochemical model (PISCES-v2) complemented with nitrogen isotopes, coupled offline with the physical model (NEMO). The authors perform 4 model experiments to assess the isolated and combined effects of the two anthropogenic perturbations under preindustrial and high emissions RCP8.5 scenario.

They find that climate change effects overwhelm the effects deriving from N deposition, and that both perturbations lead to an increase in oceanic N fluxes. With additional sensitivity simulations they show that the indirect climate change effects, via changes in circulation, lead, relative to the direct temperature effects on metabolic rates, the observed changes.

They find that these anthropogenic perturbations lead to a decline in $\delta^{15}\text{N}$ in the low-latitude twilight zone. The authors explain this finding as the effect of intensifying nitrogen limitation in the low latitude regions due to reduced nutrient supply via upwelling. They also find that declines in $\delta^{15}\text{N}$ emerge more rapidly than background variability (ToE), particularly over some well defined regions.

This study is well written, focused and addresses some important scientific questions. However, I have some concerns that refrain me from recommending its publication in its present form. The effects of atmospheric N deposition and climate change on the N cycle balance have been discussed in earlier studies. I would suggest to focus on the isotopic signature response to the direct and indirect effects of climate change, which to my knowledge has not been shown yet. In this regard, the interpretation of the declining trend in the isotopic $\delta^{15}\text{N}$ need clarification and I recommend to expand the analysis and discussion explicitly considering the role of water column denitrification and atmospheric deposition. My articulated comments and recommendations follow here below:

- The 1st part "Anthropogenic Alteration" is not really novel. Similar experiments accounting for the single (Somes et al., 2106; Yang and Gruber, 2016; Landolfi et al., 2017; Battaglia and Joos, 2018) and the combined perturbations (Moore et al., 2013; Landolfi et al., 2017) on the N cycle have been done previously, and to my surprise little comparison or acknowledgment of this previous work is carried out here. Earlier works have shown a higher degree of compensating feedbacks with little net effect on the N fluxes as compared to the results presented here. How do the used biogeochemical model characteristics (reduced expansion of oxygen minimum zones (OMZ) and implicit N_2 fixation) (Amunot et al., 2015) contribute to the small negative feedbacks and projected N accumulation projected in this study relative to earlier works should be discussed and stated more clearly, I think. Understanding sources of model mismatch is important for reducing uncertainties in model projection and a discussion on this is very welcome.

- The N source-sink balance is shifted towards N accumulation however N become more limiting at the surface. Where does this N accumulate? And what is its isotopic signal? This brings me to my next concern: $\delta^{15}\text{N}$ decline in the twilight zone– the mechanism proposed remains unclear to me. With declining nutrient availability, I would expect less fractionation such that the $\delta^{15}\text{N}$ of PON is closer to that of NO_3 . I'm not sure the 0D model used by the authors to explain the result of the more complex model can best represent the ocean low-latitude conditions. In the 0D model, which starts from high NO_3 (20 mmol/m³) initial conditions, NO_3 fractionation occurs, enriching left over $\delta^{15}\text{N}$ NO_3 and depleting $\delta^{15}\text{N}$ PON, UNTIL NO_3 becomes limiting, which occurs at around 2 mmol/m³ (given the choice of the NO_3 half-saturation constant). As NO_3 becomes limiting, fractionation should be reduced

as phytoplankton take whatever NO₃ is left over. Thus, δ¹⁵N PON depletion would be reduced as NO₃ concentrations approach limiting conditions, after which I would expect (for mass conservations) δ¹⁵N PON to increase again. This might not be visible in supplementary fig 8 given that the NO₃ range goes only down to 10 mmol/m³ ie: 10% lower than maximum growth rate, away from growth limiting conditions. I'm afraid that as NO₃ uptake proceeds from 10 to ~0 NO₃ mmol/m³, the increase in the δ¹⁵N would be visible. Thus, I'm not convinced that the proposed process can help to lower δ¹⁵N PON and δ¹⁵N NO₃ in the nutrient limited subtropical regions where NO₃ concentrations are close to detection limits.

- Figures 2 and 3 suggest that the projected reduction in water column denitrification and the increase in N inputs from atmospheric deposition can contribute to explain the modelled δ¹⁵N decline. Both processes, via their typical fractionation, affect the whole δ¹⁵N inventory with global net δ¹⁵N NO₃ change (Altabet et al., 2007). It is not clear to me as to why these inventory changing processes have not been discussed when explaining the δ¹⁵N decline. I suggest to expand and address with more detail these mechanics potentially leading to the δ¹⁵N decline.

- I very much like the ToE analysis. However, I feel that this does not go too far in its potential. If isotopes are a good tool to detect changes particularly over some specific regions, can/do observations show any of these changes similarly to what shown in the current model? I feel that a more in-depth discussion on this aspect would widen the impact of the paper. Also how do the isotopic changes projected in this study fit with/ help the interpretation of past isotopic signals?

Specific comments:

Line 25-26: "...Overall, a shift in the global nitrogen source-sink balance towards accumulation is synonymous with climate change" given that this result is in contrast with previous works this statement should be revised.

Line 50: "identify the primary drivers and response of the nitrogen cycle". The effect of N atmospheric deposition and climate change (warming) on the N cycle balance has been assessed in other studies already (see earlier comment). I would suggest to focus on the change in the isotopic signatures under climate change that, to my knowledge has not been shown before.

Line 71: "...intensifies nitrogen limitation, reduces NPP and consequently shifts the bioavailable nitrogen budget into accumulation." I don't see how intensification of the N limitation can lead to a consequent shift to N accumulation, rephrase.

Line 90-91: do all other CMIP5 ESM simulations include N deposition as well? If not then you should compare with the climate change only model runs.

Line 95: how does the decline in zooplankton grazing affect δ¹⁵N in PON? That may also contribute to the δ¹⁵N decline?

Line 98: "comparison with field investigations" this is a bit weird given that you're comparing with end-of-the 21st century model projection

Line 99-100: earlier studies have shown stronger feedback processes with (Landolfi et al., 2017) and without climate change effects (Krishnamurthy et al., 2010; Somes et al., 2016; Yang and Gruber, 2016). The expansion of OMZ and increase in water column denitrification allowed to get rid of the extra N deposited. Here about 30 Tg N/y accumulate, suggesting that a large fraction (>50%) of the N deposited from the atmosphere actually stays in the ocean. What is the cause of the mismatch with earlier estimates? reduced model sensitivity to the feedback processes? Is this associated with particular model configuration? – this should be discussed.

Line 123:124: ToE: regional variability. This is very interesting. Are the emergent regional changes in line with observations in those particular region? If so this result can guide observational efforts towards specific regions where changes are more likely to be observed.

Line 132-140: ToE: temporal variability. Are the emergent trends in line with del $^{15}\text{NO}_3$ observations?

Line 152-155 the depletion of twilight zone $\delta^{15}\text{N}$ across the tropics and subtropics (Figure 3b) appears stronger than that of $\delta^{15}\text{N}$ -depleted organic matter from the overlying euphotic zone (Figure 3a). Wouldn't this suggest that other processes act in addition to the sinking of $\delta^{15}\text{N}$ depleted organic matter? In the equatorial Pacific the decline in water column denitrification may cause the contrast between the upper ocean ^{15}N organic matter and the subsurface $\delta^{15}\text{N}$ NO_3 .

Line 162-162: I don't really understand this. The lack of del $\delta^{15}\text{N}$ PON tropical-subtropical spatial gradients do not seem to support this. With the exception of high latitudes and the eastern equatorial Pacific where the increase of NPP is associated with del $\delta^{15}\text{N}$ PON enrichment (due to fractionation during NO_3 uptake) the tropical and subtropical regions are characterized by low $\delta^{15}\text{N}$ PON (Fig. 2a). If nutrient limitation is increasing the fractionation during phytoplankton NO_3 uptake would be reduced.

Line 164: Dissolved organic nutrients with longer lifetimes penetrate into gyres, while inorganic nutrients are used near the gyre margins (Letscher et al., 2016). What is the remineralization timescale of DON used? DON fractionation can help to lower del $^{15}\text{NO}_3$, is this quantifiable?

Line 168-170: 0D If understand this correctly the model starts from high NO_3 (20 mmol/m³) initial conditions, as phytoplankton grown and take up NO_3 fractionation occurs, enriching left over d $\delta^{15}\text{N}$ NO_3 and depleting $\delta^{15}\text{N}$ PON so far so good. As NO_3 becomes limiting, around 2 mmol/m³ fractionation should be reduced and phytoplankton take whatever NO_3 is left over, for ^{15}N mass conservations ^{15}N PON is expected to increase again. This might not be visible in supplementary fig 8 given that the NO_3 range goes only down to 10 mmol/m³ ie: 10% lower than maximum growth rate, away from growth limiting conditions. I'm afraid that as NO_3 uptake proceeds from 10 to ~0 NO_3 mmol/m³, the increase in the $\delta^{15}\text{N}$ would be visible.

Line 176: ..."N limitation proximate driver." The mechanism proposed is unclear to me

Line 188-189: "Climate change effects on mixing were therefore the primary driver of .." sounds like an over simplification, "mixing" is one of the components affecting circulation, I suggest to rephrase.

Line 208: Do OMZ expand in your model experiments?

Line 209: do you account for CO_2 fertilization in your model? I think these statements are confusing as it is not clear what is

Supplementary Note I: I think there are some errors in sings the fractionation factors.

Altabet, M. A. (2007). Constraints on oceanic N balance/imbalance from sedimentary ^{15}N records. *Biogeosciences* 4, 75–86. doi: 10.5194/bg-4-75-2007.

Krishnamurthy, A., J. K. Moore, N. Mahowald, C. Luo, and C. S. Zender (2010), Impacts of atmospheric nutrient inputs on marine biogeochemistry, *J. Geophys. Res.*, 115, G01006, doi:10.1029/2009JG001115.

Landolfi, A., C. J. Somes, W. Koeve, L. M. Zamora, and A. Oschlies (2017), Oceanic nitrogen cycling

and N₂O flux perturbations in the Anthropocene, *Global Biogeochem. Cycles*, 31, 1236–1255, doi:10.1002/2017GB005633.

Moore, J. K., K. Lindsay, S. C. Doney, M. C. Long, and K. Misumi (2013), Marine ecosystem dynamics and biogeochemical cycling in the community Earth system model [CESM1(BGC)]: Comparison of the 1990s with the 2090s under the RCP4.5 and RCP8.5 scenarios, *J. Clim.*, 26(23), 9291–9312, doi:10.1175/JCLI-D-12-00566.1.

Somes, C. J., A. Landolfi, W. Koeve, and A. Oschlies (2016), Limited impact of atmospheric nitrogen deposition on marine productivity due to biogeochemical feedbacks in a global ocean model, *Geophys. Res. Lett.*, 43, 4500–4509, doi:10.1002/2016GL068335.

Yang, S., and N. Gruber (2016), The anthropogenic perturbation of the marine nitrogen cycle by atmospheric deposition: Nitrogen cycle feedbacks and the 15N Haber-Bosch effect, *Global Biogeochem. Cycles*, 30, 1418–1440, doi:10.1002/2016GB005421.

Reviewer #2:

Remarks to the Author:

This manuscript by Buchanan et al. uses the PISCES global ocean biogeochemical model with nitrogen isotopes to project changes of atmospheric nitrogen deposition and climate on changes to the $\delta^{15}\text{N}$ distribution. Their main finding is that reduced productivity driven by climate change effects is detectable through significantly decreased twilight zone $\delta^{15}\text{N}$. Their simulations also predict that climate change is increasing the N budget.

I think this is a really nice set of model experiments that a lot can be learned from and important conclusions can be made, making it a nice study for this journal. The manuscript is well written and figures clearly presented. However, I find the main interpretation of the model results somewhat unconvincing and explanation oversimplified (see comments below). On the other hand, I think the Time of Emergence analysis makes a very important point about the usefulness of $\delta^{15}\text{N}$ which is not typically included in models. I think after some clarifications and revisions this can be an excellent manuscript for publication.

Best regards,
Christopher Somes
GEOMAR Helmholtz Centre for Ocean Research Kiel

Major Comments:

lines 58-62: "In general, NPP and sinks increase $\delta^{15}\text{N}$ "
"... nitrogen isotopes to probe past changes in nitrogen cycling and NPP"

Very rarely is a direct link made with $\delta^{15}\text{N}$ and NPP, it is typically $\delta^{15}\text{N}$ and surface DIN utilization (in the absence of source/sink processes), which is a mechanism that operates much differently than NPP. For example, some of the highest $\delta^{15}\text{N}$ values occur in the low productivity gyres (as long as N₂ fixation is not occurring) due to high DIN utilization from depletion, while there is a notable minimum in $\delta^{15}\text{N}$ across the highly productive equatorial Pacific due to low utilization caused by iron limitation (e.g. [Altabet and Francois, 1994]).

The paleo studies cited (22-24) all focus on external sources N₂ fixation and denitrification, which are affected by and interact with NPP but are far from a direct proxy of NPP. Even considering a region far removed from external source/sink processes such as the glacial Southern Ocean is quite complicated,

where productivity proxies show higher productivity in the Subantarctic zones versus lower productivity in the higher latitude Antarctic zone (e.g. [Kohfeld et al., 2005]). However, the d15N signal shows a fairly consistent increase throughout both zones, which can be explained by higher utilization due to iron deposition in the SAZ and lower supply via enhanced stratification in the AZ ([Kemeny et al., 2018] [François et al., 1997; Robinson and Sigman, 2008]). I bring up this example because I worry that non-experts may think a simple relationship between NPP and d15N exists, which is not always the case and I think the highly variable nature of the surface d15N model results support this as well.

lines 161-163: "growing nitrogen limitation across the lower latitude caused more 15N to be removed from the euphotic zone in sinking organic matter and therefore depleted 15N within the remaining dissolved bioavailable nitrogen."

I'm having trouble understanding how this mechanism operates. When view spatial gradients, typically the opposite happens. Phytoplankton preferentially incorporate the 14N when it is readily available, leaving 15N-enriched DIN to advect into the gyres. For example, this is why you see a minimum of d15N-PON in the productive equatorial Pacific that trends to higher values in the more N-limited gyres ([Altabet and Francois, 1994]). I'm confused why this is not occurring in the model.

In supplementary Figure 8 (0D model), what is the delta d15N-POM reflecting? Is it total POM accumulated throughout the entire "transport" to the DIN depleted gyre or the final d15N-POM in the gyre? It is confusing since the x-axis is initial bioavailable DIN and not decreasing DIN during transport to the gyre - I'm not sure what the d15N-POM signature is when DIN is nearly fully depleted. If DIN is eventually fully depleted, the total d15N-POM produced should equal the original supply due to mass balance so I'm not sure how this continuous d15N decrease can occur.

I guess what may be happening is when you start with less DIN, the amount of DIN consumed during depletion towards the gyre is less, which reduces utilization and thus also reduces enrichment of d15N causing lower delta d15N in the different scenarios with lower initial DIN. But the spatial/temporal d15N trend from the upwelling zone to the gyre has to increase in each individual scenario as DIN is depleted because phytoplankton still first preferentially consume 14N, right? I'm not sure I'm understanding it correctly, I think it would be helpful to show the transient changes of d15N-POM as the system is becoming more DIN depleted from the upwelling zone towards the gyre, not only the steady-state results based on initial DIN.

I also wonder if applying an open system fractionation system to the 0D experiment is realistic. You noted there is loss due to sinking POM so the system is not completely closed, but the DIN source is cut-off so I think the closed system fractionation would make more sense and ensure total 15N mass conservation from d15N-DIN to d15N-POM.

I'm skeptical about to what degree the NPP mechanism in Figure 3d is actually occurring in the global model given the highly variable surface d15N signal (Figure 3a, Supp. Fig. 7a). And looking more closely, the largest area of decreased NPP is the central equatorial Pacific (Figure 1c), but there is hardly any decrease in twilight zone d15NO3 there (Figure 2a). The largest decrease of twilight zone d15NO3 occurs in the subtropical western/central North Pacific, but at the surface there is very little change and even a slight increase in NPP in the northern part of this signal. It seems to me there is a lot more going on than a simple relationship to NPP.

N2 fixation and N deposition effects

I see a much stronger spatial match of the combined increases of N2 fixation and N deposition with the decreased twilight zone delta d15NO3. I think you have a complicated surface utilization signal, which is completely expected due to all the processes that control utilization and how they all differ

regionally. Then deeper in the twilight zone, you are beginning see this noted net nitrogen accumulation of low $\delta^{15}\text{N}$ near the source increases since they introduce very low $\delta^{15}\text{N}$ into the system. Of course it is difficult for me to understand everything here, but I am surprised these processes are not considered to be significantly contributing the decreased $\delta^{15}\text{NO}_3$, although lines 210-211 briefly mention this potential effect from N_2 fixation. Have you checked if these areas of decreased twilight zone $\delta^{15}\text{NO}_3$ are associated with nitrogen accumulation?

My past modeling has demonstrated that even small changes in N_2 fixation in the N deprived gyres can have large impacts on $\delta^{15}\text{N}$ since there is little to begin with, e.g. shown by modifying their iron limitation in the modern [Somes et al., 2010] and LGM [Somes et al., 2017](please don't feel obligated to cite these, I mention them to support my statements on the importance of N_2 fixation), and I think N deposition would have a similar importance where rates are high.

Time of Emergence

As mentioned above, I think this is a fantastic point to make that will be very important to the broader marine biogeochemistry community since it is often so hard to quantify trends with noisy and sparse rate measurements.

Other Literature

Please replace the [Somes et al., 2016] citation with [Landolfi et al., 2017], which is much more relevant here since the latter study also includes warming and N deposition individually and combined, very similar to here (albeit without isotopes). I think it would be useful to have one paragraph noting key differences with the Landolfi et al (2017) and Yang and Gruber (2016) studies in the main text, but I leave this decision up to you since space may be tight.

For example, the negative feedback of N fixation in response to N deposition is much weaker in your model, allowing increased N_2 fixation and more net nitrogen accumulation to occur in the climate+N deposition scenario compared to Landolfi et al (2017), although our model did predict a slightly higher N inventory in the combined simulation. I would be interested more generally on what is driving this increase in N_2 fixation, but perhaps that would take up too much space since the focus is on N isotopes, so again this is your decision. The Yang and Gruber (2016) study seems to show a stronger effect from N deposition alone on $\delta^{15}\text{N}$ than your model (at least that is the impression I get from reading their paper compared to yours), I think it would also be useful to comment on this.

Figures

The figures are well presented. But I think one reason contributing to my difficulty to understand this proposed mechanism is every single figure shows $\delta^{15}\text{N}$. Especially since this is the first use of nitrogen isotopes in the PISCES model, I think the initial preindustrial spatial $\delta^{15}\text{N}$ distributions that the projected changes arise from should also be shown in the supplementary material, not only statistical metrics.

Concluding Remarks

I think this is a really interesting study, especially with the important point of the Time of Emergence. However it is still important to get the mechanism(s) correct. Even if I am missing and/or not understanding something correctly and you are convinced about your general interpretation/mechanism, I still think it should be mainly explained in terms of changes to surface DIN utilization, rather than directly relating it to NPP. And to be fair, fractionation and N limitation are briefly mentioned in the explanation, but the focus on NPP in the introduction and conclusion will give too much of an oversimplified view in my opinion to the more general scientific audience.

Altabet, M. A., and R. Francois (1994), Sedimentary nitrogen isotopic ratio as a recorder for surface

ocean nitrate utilization, *Global Biogeochem. Cycles*, 8(1), 103-116, doi: 10.1029/93gb03396.

François, R., M. A. Altabet, E.-F. Yu, D. M. Sigman, M. P. Bacon, M. Frank, G. Bohrmann, G. Bareille, and L. D. Labeyrie (1997), Contribution of Southern Ocean surface-water stratification to low atmospheric CO₂ concentrations during the last glacial period, *Nature*, 389(6654), 929-935, doi: 10.1038/40073.

Kemeny, P. C., E. R. Kast, M. P. Hain, S. E. Fawcett, F. Fripiat, A. S. Studer, A. Martínez-García, G. H. Haug, and D. M. Sigman (2018), A Seasonal Model of Nitrogen Isotopes in the Ice Age Antarctic Zone: Support for Weakening of the Southern Ocean Upper Overturning Cell, *Paleoceanography and Paleoclimatology*, 33(12), 1453-1471, doi: 10.1029/2018pa003478.

Kohfeld, K. E., C. Le Quere, S. P. Harrison, and R. F. Anderson (2005), Role of marine biology in glacial-interglacial CO₂ cycles, *Science*, 308(5718), 74-78, doi: 10.1126/science.1105375.

Landolfi, A., C. J. Somes, W. Koeve, L. M. Zamora, and A. Oschlies (2017), Oceanic nitrogen cycling and N₂O flux perturbations in the Anthropocene, *Global Biogeochemical Cycles*, 31(8), 1236-1255, doi: 10.1002/2017gb005633.

Robinson, R. S., and D. M. Sigman (2008), Nitrogen isotopic evidence for a poleward decrease in surface nitrate within the ice age Antarctic, *Quaternary Science Reviews*, 27(9-10), 1076-1090, doi: 10.1016/j.quascirev.2008.02.005.

Somes, C. J., A. Schmittner, and M. A. Altabet (2010), Nitrogen isotope simulations show the importance of atmospheric iron deposition for nitrogen fixation across the Pacific Ocean, *Geophys. Res. Lett.*, 37(23), L23605, doi: 10.1029/2010gl044537.

Somes, C. J., A. Landolfi, W. Koeve, and A. Oschlies (2016), Limited impact of atmospheric nitrogen deposition on marine productivity due to biogeochemical feedbacks in a global ocean model, *Geophysical Research Letters*, 43, 4500-4509, doi: 10.1002/2016gl068335.

Somes, C. J., A. Schmittner, J. Muglia, and A. Oschlies (2017), A three-dimensional model of the marine nitrogen cycle during the Last Glacial Maximum constrained by sedimentary isotopes, *Frontiers in Marine Science*, 4(108), doi: 10.3389/fmars.2017.00108.

Reviewer #3:

Remarks to the Author:

In this study, the authors model the impact of climate change and anthropogenically-driven increases in atmospheric nitrogen deposition on net primary production and the accumulation of fixed nitrogen in the ocean. Surprisingly (to me), they found that the RCP8.5 climate change scenario led to a larger change in the nitrogen inventory than the nitrogen deposition flux. Specifically, climate change led to an accumulation of nitrogen in the ocean through enhanced stratification, decreasing nutrient delivery to the euphotic zone and driving a decrease in NPP. The authors use a novel approach based on N isotope patterns in nitrate and particulate organic matter to address an important question about the future evolution of the marine N cycle. I have some questions for further clarification of their methods, results, and discussion.

Lines 67-70: I thought it would be helpful if these references to 'climate change' included some of the more specific drivers of the phenomenon being assessed. It's difficult to compare here the impact of a complex suite of environmental changes due to 'climate change' with a very specific quantifiable process such as anthropogenic increases in N deposition.

Lines 157-165: This section was a little counter-intuitive to me and could probably be clarified. Especially with line 162, where increasing N utilization should lead to more export of ^{15}N , and a lower concentration of ^{15}N in the euphotic zone, but I would expect a higher $^{15}\text{N}/^{14}\text{N}$ ratio. I'm confused here about the discussion of the drivers of ^{15}N concentration without consideration of the isotope ratio that should drive the $\delta^{15}\text{N}$ values. In this same section, I would expect that a nutrient redistribution driven by climate change could lead to a redistribution of N isotopes, but it should result in increases in some regions to compensate the decreases in others. Overall, if there is no new N added, an isotope mass balance should be conserved. I think this discussion would benefit from this perspective of a redistribution of N isotopes and discuss where corresponding increases in $\delta^{15}\text{N}$ occur that offset the decreases in the subtropical thermocline.

Lines 178-180: I think it would have helped for this description to come earlier. In many of the earlier references to 'climate change', I wondered what mechanism was coming into play.

Lines 396-398: What is meant by 'mean conditions' in these contexts?

Lines 431-432: A little more explanation would have helped in this section. Specifically, what is meant by 'annually-average values' and 'normalized using the linear slope and mean of the preindustrial control experiment'? What features from the preindustrial control time series were 'centered on zero'?

Line 490: What is the origin of the $(1 - N_{\text{lim}} * ENPP / 1000)$ term in this equation?

Lines 493-500: Why was a quadratic term chosen to describe respiration? Can you please define and give units for the terms in the equations provided here? I don't understand why you can add respiration and mortality in the last equation since they don't appear to have the same units in the prior equations, with mortality linearly dependent on PON and respiration scaling with PON squared.

Lines 505-506: These equations don't appear to have the same units as N_{exported} has an extra PON term in it that does not exist in N_{recycled} .

Supplementary material:

Note 1: It is more typical for a fractionation factor to be represented by alpha (α) a ratio of isotope ratios, rather than epsilon (ϵ), which is generally given in per mil units.

Reviewer 1:

In this study the authors want to assess the effect and relative contribution of atmospheric N deposition and climate change on the oceanic nitrogen cycle and on the resulting $\delta^{15}\text{N}$ isotopic signature over the 21st century.

The tool used is a global biogeochemical model (PISCES-v2) complemented with nitrogen isotopes, coupled offline with the physical model (NEMO). The authors perform 4 model experiments to assess the isolated and combined effects of the two anthropogenic perturbations under preindustrial and high emissions RCP8.5 scenario.

They find that climate change effects overwhelm the effects deriving from N deposition, and that both perturbations lead to an increase in oceanic N fluxes. With additional sensitivity simulations they show that the indirect climate change effects, via changes in circulation, lead, relative to the direct temperature effects on metabolic rates, the observed changes.

They find that these anthropogenic perturbations lead to a decline in $\delta^{15}\text{N}$ in the low-latitude twilight zone. The authors explain this finding as the effect of intensifying nitrogen limitation in the low latitude regions due to reduced nutrient supply via upwelling. They also find that declines in $\delta^{15}\text{N}$ emerge more rapidly than background variability (ToE), particularly over some well defined regions.

This study is well written, focused and addresses some important scientific questions. However, I have some concerns that refrain me from recommending its publication in its present form. The effects of atmospheric N deposition and climate change on the N cycle balance have been discussed in earlier studies. I would suggest to focus on the isotopic signature response to the direct and indirect effects of climate change, which to my knowledge has not been shown yet. In this regard, the interpretation of the declining trend in the isotopic $\delta^{15}\text{N}$ need clarification and I recommend to expand the analysis and discussion explicitly considering the role of water column denitrification and atmospheric deposition. My articulated comments and recommendations follow here below:

We want to thank the reviewer for their positive and useful comments. They, like the other reviewers, have asked for a clearer explanation of the mechanism causing the $\delta^{15}\text{N}$ declines and have asked us to acknowledge a few prior studies on this topic. These are acknowledged and addressed in our responses below.

- The 1st part “Anthropogenic Alteration” is not really novel. Similar experiments accounting for the single (Somes et al., 2106; Yang and Gruber, 2016; Landolfi et al., 2017; Battaglia and Joos, 2018) and the combined perturbations (Moore et al., 2013; Landolfi et al., 2017) on the N cycle have been done previously, and to my surprise little comparison or acknowledgment of this previous work is carried out here. Earlier works have shown a higher degree of compensating feedbacks with little net effect on the N fluxes as compared to the results presented here. How do the used biogeochemical model characteristics (reduced expansion of oxygen minimum zones (OMZ) and implicit N_2 fixation) (Amunot et al., 2015) contribute to the small negative feedbacks and projected N accumulation projected in this study relative to earlier works should be discussed and stated more clearly, I think. Understanding sources of model mismatch is important for reducing uncertainties in model projection and a discussion on this is very welcome.

We agree with the reviewer that an extended discussion of earlier studies around the changing nitrogen inventory is needed, which in turn warrants acknowledgement of model uncertainty. We

have addressed this with new text within the section “Anthropogenic alteration” and in the discussion:

Lines 97 – 100: *“New nitrogen supply from biological nitrogen fixation declined from 78.8 to 73.9 Tg N yr⁻¹ (-6% of its preindustrial rate) by 2081-2100, consistent with other ESM simulations that considered increasing nitrogen deposition^{3,31-35}, and displayed a clear tropical to subtropical shift (Figure 1b).”*

Lines 108 – 110: *“While these gains in the global marine nitrogen budget are greater than those reported previously³, a common inter-model response to anthropogenic impacts appears to be a shift towards nitrogen accumulation.”*

As the reviewer is likely aware, all biogeochemical models suffer from biases in their oxygen fields and other biogeochemical properties that add to uncertainties in the integrated response of the N cycle. A multi-model comparison of physical and biogeochemical architecture at the root of these biases is, however, outside the scope of this work. However, this issue is important and we are glad to draw attention to it in this study:

Lines 267 – 276: *“A primary dependency on organic matter flux is reflected in our experiments, where nitrogen sinks declined despite a slight increase in hypoxic ($O_2 < 80 \text{ mmol m}^{-3}$) water volume of 0.6% by 2081-2100, relative to preindustrial conditions. If nitrogen sinks do decline as less organic matter sinks out of the euphotic zone, then our results suggest that the bioavailable nitrogen reservoir may accumulate (Figure 1g). While the response of hypoxic zones, and hence nitrogen sinks, to climate change is subject to considerable uncertainty⁴⁸, the shift towards accumulation is consistent with previous modelling³ and may be reinforced by including currently unrepresented processes, such as the stimulation of nitrogen fixation by increasing pCO_2 ⁴⁶ and anthropogenic iron deposition⁵⁰. Net accumulation of bioavailable nitrogen in the 21st century, if realised, ...”*

- The N source-sink balance is shifted towards N accumulation however N become more limiting at the surface. Where does this N accumulate? And what is its isotopic signal?

Comparing with the preindustrial experiment, N accumulates in the Southern Ocean, between 20-40° in the mesopelagic of each hemisphere, and in the deep North Atlantic as the AMOC slows down. It declines in the surface basically everywhere north of 50°S, from 0-2000 m in the tropics, from 0-2000 metres in the southern mid-latitudes and in the Arctic Ocean.

There is however no clear link with the $\delta^{15}N$ trends in the twilight zone. We show below the lack of correlation between changes in twilight $\delta^{15}N_{NO_3}$ and NO_3 concentrations. Therefore, whether NO_3 accumulates or decreases in a region of the low latitude twilight zone appears to have little

relationship with the $\delta^{15}\text{NNO}_3$ trends.

$\delta^{15}\text{N-NO}_3$ change at 300 metres (climate change only experiment)

NO_3 change at 300 metres (climate change only experiment)

This brings me to my next concern: $\delta^{15}\text{N}$ decline in the twilight zone— the mechanism proposed remains unclear to me. With declining nutrient availability, I would expect less fractionation such that the $\delta^{15}\text{N}$ of PON is closer to that of NO_3 . I'm not sure the OD model used by the authors to explain the result of the more complex model can best represent the ocean low-latitude conditions. In the OD model, which starts from high NO_3 (20 mmol/m³) initial conditions, NO_3 fractionation occurs, enriching left over $\delta^{15}\text{N}$ NO_3 and depleting $\delta^{15}\text{N}$ PON, UNTIL NO_3 becomes limiting, which occurs at around 2 mmol/m³ (given the choice of the NO_3 half-saturation constant). As NO_3 becomes limiting, fractionation should be reduced as phytoplankton take whatever NO_3 is left over. Thus, $\delta^{15}\text{N}$ PON depletion would be reduced as NO_3 concentrations approach limiting conditions, after which I would expect (for mass conservations) $\delta^{15}\text{N}$ PON to increase again. This might not be visible in supplementary fig 8 given that the NO_3 range goes only down to 10 mmol/m³ ie: 10% lower than maximum growth rate, away from growth limiting conditions. I'm afraid that as NO_3 uptake proceeds from 10 to ~ 0 NO_3 mmol/m³, the increase in the $\delta^{15}\text{N}$ would be visible. Thus, I'm not convinced that the proposed process can help to lower $\delta^{15}\text{N}$ PON and $\delta^{15}\text{N}$ NO_3 in the nutrient limited subtropical regions where NO_3 concentrations are close to detection limits.

We agree that the mechanism could be more clearly explained. The revised manuscript has improved both the schematic of Figure 3d by adding numbers that are used in the text and caption for a clearer explanation, and has split section 3 into two sections (3 and 4) with revised text. These revisions hopefully have improved the clarity of the mechanism.

Section 3 (lines 161 – 227) now reads:

“While low $\delta^{15}\text{N}$ from nitrogen deposition was important for 20th century declines in $\delta^{15}\text{N}_{\text{NO}_3}$, consistent with previous modelling³⁴, climate-driven declines in the 21st century potentially involved changes to numerous nitrogen cycle processes, the individual effects of which are difficult to isolate. Potential contributors included a decrease in phytoplankton production and fractionation due to nitrogen limitation, an increase in nitrogen fixation, an increase in zooplankton recycling, a decrease in denitrification, or a physical redistribution of the dissolved nitrogen compounds by a changing circulation (or any combination of these processes). Here, we demonstrate that changes ultimately linked to increasing nitrogen limitation, namely a decrease in phytoplankton production and fractionation potentially supplemented by a tropical-subtropical shift of nitrogen fixation, was the primary driver of the broad, simulated $\delta^{15}\text{N}_{\text{NO}_3}$ declines that emerged in the twilight zones of the low latitude oceans.

Firstly, we infer that ^{15}N -depleted organic matter in the euphotic zone drove the $\delta^{15}\text{N}_{\text{NO}_3}$ declines in the twilight zone. Widespread declines in $\delta^{15}\text{N}_{\text{POM}}$ (Figure 3a) delivered less ^{15}N to the twilight zone via the remineralisation of sinking organic material. This inference was further supported by an analysis of purely biogeochemical $^{15}\text{NO}_3$ fluxes in the twilight zone, which disentangled local biogeochemical effects from circulation effects (see Methods), and was able to reproduce the depletion of $\delta^{15}\text{N}_{\text{NO}_3}$ across the tropics and subtropics (Figure 3b). An exception were regions influenced by Southern Ocean mode waters in the South Pacific and Indian oceans⁴³ where upstream increases in NPP and nitrogen consumption in response to climate change (Figure 1c) led to a biogeochemically driven enrichment in $\delta^{15}\text{N}_{\text{NO}_3}$. However, the broad biogeochemical tendency was to lower twilight zone $\delta^{15}\text{N}_{\text{NO}_3}$ and this was linked to depleted $\delta^{15}\text{N}_{\text{POM}}$ sinking out of subtropical euphotic zones.

Secondly, the decline in nitrogen assimilation rates by phytoplankton (by 297 Tg N yr⁻¹ globally; Figure 1c) far exceeded parallel changes in the rate of sources and sinks, and made an important contribution to the simulated isotopic declines. Increasing nitrogen limitation of phytoplankton was responsible for reduced nitrogen assimilation across the lower latitude oceans during the 21st century in our model, with bioavailable nitrogen declining by 0.5-4 mmol m⁻³ (5-40%) in the tropical Pacific and Atlantic euphotic zones (by 2081-2100 relative to 1986-2005; Figure 3c). These trends are consistent with the common projections of declining upper ocean nitrate inventories across ESM studies that are largest in the low latitude upwelling systems^{10,11}. This is important because changing nitrogen availability not only affects the success of nitrogen fixers, but also affects the rate of nitrogen uptake by phytoplankton and the degree of nitrogen isotope fractionation, which combine to control the horizontal $\delta^{15}\text{N}_{\text{NO}_3}$ gradients across the low latitude ocean^{1,20,22,23}. Increasing nitrogen limitation under climate change therefore led to both a stimulation of nitrogen fixation and a weaker isotopic enrichment associated with phytoplankton assimilation. Weaker fractionation by phytoplankton (i.e. a weaker preference for ¹⁴N) meant that more ¹⁵N was used to create organic matter in increasingly nitrogen-limited upwelling regions, and was subsequently lost via sinking (step 1 in Figure 3d). Consequently, euphotic zone $\delta^{15}\text{N}_{\text{NO}_3}$ and $\delta^{15}\text{N}_{\text{POM}}$ values declined at the boundary between the nitrogen-replete and nitrogen-deplete regimes (step 2 in Figure 3d). Because subtropical gyres receive most of their nutrients from lateral transport²⁴, ¹⁵N-depleted nitrogen was then swept into the subtropical gyres where local nitrogen recycling and organic matter formation proceeded with lighter isotopic signatures (i.e. relatively more ¹⁴N; step 3 in Figure 3d), and, combined with local increases in biological nitrogen fixation, delivered lower $\delta^{15}\text{N}_{\text{POM}}$ values to the subtropical twilight zone.

The removal of ¹⁵N by nitrogen limited phytoplankton in the tropics followed by the transfer of ¹⁵N-depleted water to the gyres is also supported by idealised modelling. We constructed a zero-dimensional (0D) model that follows nitrogen uptake and fractionation by phytoplankton in a water parcel (see Methods; Supplementary Table 1). In this simple model, an inorganic nitrogen pool representing initial upwelled nitrogen is steadily assimilated by phytoplankton to create organic matter, and this organic matter is either remineralised back to inorganic nitrogen or removed permanently via export. For every 10% decline in initial upwelled nitrogen supplied to the water parcel (corresponding to a ~10% loss in integrated NPP during the water parcel's lifetime), there is a $0.19 \pm 0.03\%$ decline in the final $\delta^{15}\text{N}_{\text{POM}}$ produced within the gyre once nitrogen is depleted to limiting concentrations. The uncertainty of $\pm 0.03\%$ is associated with temperature changes of $\pm 4^\circ\text{C}$, suggesting that warming also plays a role by modulating growth rates and recycling (Supplementary Figure 9). Extrapolating this 'rule of thumb' to the global model, we expect $\delta^{15}\text{N}_{\text{POM}}$ declines of 0.1-0.8‰ for bioavailable nitrogen declines of 5-40%, which agrees broadly with the results of the full model by 2081-2100 (Figure 2c). Furthermore, the local enrichment of $\delta^{15}\text{N}_{\text{POM}}$ in the upwelling region in both our 0D model (compare solid and dashed lines in Figure 3d) and in the tropical Pacific in our global model (Figure 3a) clearly signifies the existence of this mechanism, where more ¹⁵N was removed from upwelling zones due to nitrogen limitation of phytoplankton (note that this feature was absent in the Atlantic due to local declines in denitrification (Figure 1g)). Ultimately, the biogeochemical consequences of increasing nitrogen limitation appeared to be the primary cause of the widespread isotopic declines."

Section 4 (lines 229 – 247) now reads:

“We examined the direct (i.e. warming on biogeochemical rates) and indirect effects (i.e. circulation altering nutrient supply) of climate change in two additional experiments. First, warming was imposed on biogeochemical processes under otherwise preindustrial conditions to mimic its effect on rates. Second, preindustrial temperatures were imposed while climate change altered the circulation to mimic its effect on substrate availability (see Methods). Direct effects of warming on biogeochemical processes showed a limited ability to reproduce the full suite of climate-driven trends, with the exception of the poleward shift in nitrogen fixers (Figure 4). In contrast, changes to ocean circulation (e.g. stratification) in the indirect effect simulation replicated changes in euphotic zone nitrate (Spearman’s rank correlation; $r_s = 0.99$), twilight zone $\delta^{15}\text{NNO}_3$ ($r_s = 0.94$), twilight zone $\delta^{15}\text{NPOM}$ ($r_s = 0.95$), NPP ($r_s = 0.69$), zooplankton grazing ($r_s = 0.69$) and water column denitrification ($r_s = 0.88$) and sedimentary denitrification ($r_s = 0.90$) seen in the full model (Figure 4). Moderate agreement for nitrogen fixation in both the direct effect ($r_s = 0.65$) and indirect effect simulations ($r_s = 0.61$) suggested that both warming and circulation changes were equally important in determining its full response, but that nitrogen fixation by itself was not sufficient to force the isotopic trends associated with climate change. The effect of climate change on biogeochemical processes via altering substrate availability was therefore a major cause of 21st century increases in nitrogen limitation, resultant NPP declines, shifts in the sources and sinks of nitrogen, and the accompanying trends in nitrogen isotopes that fingerprint the response of the nitrogen cycle.”

New schematic (change is the numbers for aid in explanation):

The concerns raised about the OD model are acknowledged and addressed. First, we have clarified what Supplementary Figure 8 (now Supplementary Figure 9) is showing in the caption, which may have caused some misunderstanding (the x axis is initial N availability and the y axis is final $\delta^{15}\text{N}_{\text{POM}}$ signature after running the OD model to equilibrium, at which point N has been fully utilised).

Finally, our OD model is constructed to conserve mass of all its tracers and this can be investigated freely by downloading the python code at https://github.com/pearseb/PISCESiso_Ncycle_analysis.

- Figures 2 and 3 suggest that the projected reduction in water column denitrification and the increase in N inputs from atmospheric deposition can contribute to explain the modelled $\delta^{15}\text{N}$ decline. Both processes, via their typical fractionation, affect the whole ^{15}N inventory with global net $\delta^{15}\text{N}$ NO_3 change (Altabet et al., 2007). It is not clear to me as to why these inventory changing

processes have not been discussed when explaining the $\delta^{15}\text{N}$ decline. I suggest to expand and address with more detail these mechanics potentially leading to the $\delta^{15}\text{N}$ decline.

It is possible that there are alternative contributions, such as denitrification, N_2 fixation and the redistribution of isotopes by the circulation. We therefore agree that these alternatives must be addressed, and we have addressed them in our revisions over section 3 and the new section 4, which are shown above.

- I very much like the ToE analysis. However, I feel that this does not go too far in its potential. If isotopes are a good tool to detect changes particularly over some specific regions, can/do observations show any of these changes similarly to what shown in the current model? I feel that a more in-depth discussion on this aspect would widen the impact of the paper. Also how do the isotopic changes projected in this study fit with/ help the interpretation of past isotopic signals?

We agree that this is an important part of the paper. We have therefore made it clearer that repeat hydrographic occupations are needed for $\delta^{15}\text{N}_{\text{NO}_3}$ but are currently lacking.

Lines 257 – 261: *“While repeat $\delta^{15}\text{N}_{\text{NO}_3}$ measurements along hydrographic lines are currently lacking, future occupations may detect the effects of climate change. This is because the currently available measurements of $\delta^{15}\text{N}_{\text{NO}_3}$ in the twilight zones of the Pacific ($N = 1,481$) and Atlantic ($N = 890$) have been made near peak rates of nitrogen deposition (median age of data = 2008)²⁰. Since future deposition is expected to plateau^{4,7}, any future trends should be dominated by climate-driven signals.”*

It is beyond the scope of this study to reinterpret the palaeoceanographic $\delta^{15}\text{N}$ records. However, the reviewer is correct that as that palaeoceanographers now routinely use planktonic foraminifera, some of which are sub-euphotic, as the gold-standard for reconstructing $\delta^{15}\text{N}$ records, there is an opportunity to investigate changes in low latitude nitrogen availability using sedimentary archives. We have added new text to point out this possibility:

Lines 286 - 290: *“The relative stability of the twilight zone is demonstrated by the low variability of modern-day $\delta^{15}\text{N}$ recorded in sub-euphotic foraminifera⁵⁵. As foraminifera are now routinely used to reconstruct $\delta^{15}\text{N}$ for studies of the past oceanic nitrogen cycle^{21,25,27}, our results suggest that sedimentary archives of sub-euphotic species may provide an opportunity to investigate past variations in nitrogen availability across the low latitudes.”*

Specific comments:

Line 25-26: “...Overall, a shift in the global nitrogen source-sink balance towards accumulation is synonymous with climate change” given that this result is in contrast with previous works this statement should be revised.

This point has been addressed in our previous response to the earlier reviewer comment about acknowledging prior work.

Line 50: “identify the primary drivers and response of the nitrogen cycle”. The effect of N

atmospheric deposition and climate change (warming) on the N cycle balance has been assessed in other studies already (see earlier comment). I would suggest to focus on the change in the isotopic signatures under climate change that, to my knowledge has not been shown before.

We agree that a clear novelty of our work is in the isotopic trends. However, we have maintained some focus (first section “Anthropogenic alterations”) on how the N cycle is changing in order to talk about the $\delta^{15}\text{N}$ trends, which rely on an understanding of N cycle changes. Moreover, we acknowledge the prior work in our revisions as requested by the reviewer.

Line 71: ...”intensifies nitrogen limitation, reduces NPP and consequently shifts the bioavailable nitrogen budget into accumulation.” I don’t see how intensification of the N limitation can lead to a consequent shift to N accumulation, rephrase.

We have made the links clearer.

Lines 75 – 78: *“We find that nitrogen isotopes fingerprint the dominant role of climate change, specifically how circulation changes can intensify nitrogen limitation of lower latitude ecosystems, leading to decreased NPP and nitrogen sinks, and subsequently alter the bioavailable nitrogen budget.”*

Line 90-91: do all other CMIP5 ESM simulations include N deposition as well? If not then you should compare with the climate change only model runs.

They do not and because of this we have compared with the Landolfi and Moore papers instead.

Line 95: how does the decline in zooplankton grazing affect $\delta^{15}\text{N}$ in PON? That may also contribute to the $\delta^{15}\text{N}$ decline?

We agree that zooplankton could play a role and we have acknowledged and addressed this point in the paper by adding the following text.

Lines 165 – 168: *“Potential contributors included a decrease in phytoplankton production and fractionation due to nitrogen limitation, an increase in nitrogen fixation, an increase in zooplankton recycling, a decrease in denitrification, or a physical redistribution of the dissolved nitrogen compounds by a changing circulation (or any combination of these processes).”*

Line 98: “comparison with field investigations” this is a bit weird given that you’re comparing with end-of the 21st century model projection

The phrase “consistent with field investigations” refers to the Kavelage study where N loss processes in the Eastern Tropical South Pacific were driven by variations in organic matter flux, which make sense given that these processes are carried out by heterotrophic bacteria/archaea.

To make the dependency on organic matter rain more obvious we have slightly altered this sentence to read:

Lines 102 – 106: *“Declines in nitrogen utilisation by zooplankton grazing (Figure 1d) were consistent with trophic amplification³⁸, while denitrification changes (in both the water column and sediments; Figure 1e,f) depended on local changes in particulate organic matter export, a dependency consistent with field investigations³⁹ and data-constrained modelling^{35,40}.”*

Line 99-100: earlier studies have shown stronger feedback processes with (Landolfi et al., 2017) and without climate change effects (Krishnamurthy et al., 2010; Somes et al., 2016; Yang and Gruber, 2016). The expansion of OMZ and increase in water column denitrification allowed to get rid of the extra N deposited. Here about 30 Tg N/y accumulate, suggesting that a large fraction (>50%) of the N deposited from the atmosphere actually stays in the ocean. What is the cause of the mismatch with earlier estimates? reduced model sensitivity to the feedback processes? Is this associated with particular model configuration? – this should be discussed.

The reviewer is correct in that almost 30 Tg N yr⁻¹ accumulates in our combined anthropogenic experiment, but we also show that this accumulation is driven by climate change, not the increase in N deposition.

We have expanded our discussion of these effects in the first section to make this clearer.

Lines 112 - 123: *“An important point is that climate change dominated the alteration of the marine nitrogen budget. By 2081-2100, climate change had increased the bioavailable nitrogen budget by 23.7 Tg N yr⁻¹ in the absence of historical and future increases in nitrogen deposition. This increase is explained by an increase in nitrogen fixation (+7.0 Tg N yr⁻¹) and a decrease in sinks (denitrification (-13.6 Tg N yr⁻¹) and burial (-3.1 Tg N yr⁻¹)) (Supplementary Figure 5). In contrast, the anthropogenic increase in nitrogen deposition without climate change led to a small change in the budget of +4.9 Tg N yr⁻¹ (Figure 1g) due to strong compensatory feedbacks, consistent with other ESMs³²⁻³⁴, wherein newly deposited nitrogen either replaced nitrogen previously provided by nitrogen fixation (-12.0 Tg N yr⁻¹), or was rapidly removed by a local acceleration of denitrification (+6.4 Tg N yr⁻¹) and burial (+2.8 Tg N yr⁻¹; Supplementary Figure 6). The individual effects of climate change (+23.7 Tg N yr⁻¹) and nitrogen deposition (+4.9 Tg N yr⁻¹), while not perfectly additive, combined to cause the net accumulation of nitrogen in the ocean (+27.5 Tg N yr⁻¹).”*

Line 123:124: ToE: regional variability. This is very interesting. Are the emergent regional changes in line with observations in those particular region? If so this result can guide observational efforts towards specific regions where changes are more likely to be observed.

In the northwest Pacific, the emergence of d15N changes due to atmospheric deposition has been observed (Ren et al. 2017) and this is also one of the first places where our ToE analysis shows emergence (mid 20th century). However, the effects of climate change for driving anomalous d15N trends are largely emergent in the coming decades (Figure 2h) and given the lack of repeat hydrographic occupations with d15N sampling we cannot comment on d15N trends in the observations.

Lines 152 – 154: *“In the northwest Pacific, a strong decline in $\delta^{15}N_{POM}$ recorded in corals has been attributed to the rise in nitrogen deposition in recent decades³⁰, and our simulations support this attribution.”*

Line 132-140: ToE: temporal variability. Are the emergent trends in line with del 15NO3 observations?

The current global compilation of $\delta^{15}\text{N}_{\text{NO}_3}$ measurements reflects single hydrographic section occupations. As such, trends are not yet possible to extract from the data, but the coverage is sufficient to represent a baseline from which trends can be detected.

Lines 257 – 261: *“While repeat $\delta^{15}\text{N}_{\text{NO}_3}$ measurements along hydrographic lines are currently lacking, future occupations may detect the effects of climate change. This is because the currently available measurements of $\delta^{15}\text{N}_{\text{NO}_3}$ in the twilight zones of the Pacific ($N = 1,481$) and Atlantic ($N = 890$) have been made near peak rates of nitrogen deposition (median age of data = 2008)²⁰. Since future deposition is expected to plateau^{4,7}, any future trends should be dominated by climate-driven signals.”*

Line 152-155 the depletion of twilight zone $\delta^{15}\text{N}$ across the tropics and subtropics (Figure 3b) appears stronger than that of $\delta^{15}\text{N}$ -depleted organic matter from the overlying euphotic zone (Figure 3a). Wouldn't this suggest that other processes act in addition to the sinking of $\delta^{15}\text{N}$ -depleted organic matter? In the equatorial Pacific the decline in water column denitrification may cause the contrast between the upper ocean $\delta^{15}\text{N}$ organic matter and the subsurface $\delta^{15}\text{N}$ NO_3 .

As the reviewer states, alternative processes can contribute to the trends. Our extended explanation of the mechanism and our individual analyses in sections 3 and 4 acknowledge the potential role of alternative processes.

Line 162-162: I don't really understand this. The lack of $\delta^{15}\text{N}$ PON tropical-subtropical spatial gradients do not seem to support this. With the exception of high latitudes and the eastern equatorial Pacific where the increase of NPP is associated with $\delta^{15}\text{N}$ PON enrichment (due to fractionation during NO_3 uptake) the tropical and subtropical regions are characterized by low $\delta^{15}\text{N}$ PON (Fig. 2a) (We think the reviewer here means Figure 3a). If nutrient limitation is increasing the fractionation during phytoplankton NO_3 uptake would be reduced.

This has been explained above more fully, and we have provided a more thorough explanation in the revised manuscript.

Line 164: Dissolved organic nutrients with longer lifetimes penetrate into gyres, while inorganic nutrients are used near the gyre margins (Letscher et al., 2016). What is the remineralization timescale of DON used? DON fractionation can help to lower $\delta^{15}\text{N}_{\text{NO}_3}$, is this quantifiable?

There is no fractionation associated with the production or consumption of dissolved organic matter and its influence on the $\delta^{15}\text{N}_{\text{POM}}$ and $\delta^{15}\text{N}_{\text{NO}_3}$ will therefore simply be through enabling increased transport into the gyres.

Line 168-170: OD If understand this correctly the model starts from high NO_3 (20 mmol/m³) initial conditions, as phytoplankton grown and take up NO_3 fractionation occurs, enriching left over $\delta^{15}\text{N}$ NO_3 and depleting $\delta^{15}\text{N}$ PON so far so good. As NO_3 becomes limiting, around 2 mmol/m³ fractionation should be reduced and phytoplankton take whatever NO_3 is left over, for $\delta^{15}\text{N}$ mass conservations $\delta^{15}\text{N}$ PON is expected to increase again. This might not be visible in supplementary fig 8 given that the NO_3 range goes only down to 10 mmol/m³ ie: 10% lower than maximum growth rate, away from growth limiting conditions. I'm afraid that as NO_3 uptake proceeds from 10 to ~0 NO_3 mmol/m³, the increase in the $\delta^{15}\text{N}$ would be visible.

This has been addressed in earlier responses.

Line 176: ...”N limitation proximate driver.” The mechanism proposed is unclear to me This has been addressed in earlier responses.

Line 188-189: “Climate change effects on mixing were therefore the primary driver of ..” sounds like an over simplification, “mixing” is one of the components affecting circulation, I suggest to rephrase.

Agreed. We have altered “mixing” to “circulation”.

Line 208: Do OMZ expand in your model experiments?

The percentage of ocean volume hypoxia ($O_2 < 80 \text{ mmol m}^{-3}$) does increase under climate change.

We have added this information into the discussion in support of our argument that organic matter rain is most important for controlling the rate of N loss in the ocean:

Lines 267 – 269: “A primary dependency on organic matter flux is reflected in our experiments, where nitrogen sinks declined despite a slight increase in hypoxic ($O_2 < 80 \text{ mmol m}^{-3}$) water volume of 0.6% by 2081-2100, relative to preindustrial conditions.”

Line 209: do you account for CO2 fertilization in your model? I think these statements are confusing as it is not clear what is

No, we do not account for this feedback. We have clarified that this is not accounted for in our discussion by altering the sentences to:

Lines 270 – 276: *“If nitrogen sinks do decline as less organic matter sinks out of the euphotic zone, then our results suggest that the bioavailable nitrogen reservoir may accumulate (Figure 1g). While the response of hypoxic zones, and hence nitrogen sinks, to climate change is subject to considerable uncertainty⁴⁸, the shift towards accumulation is consistent with previous modelling³ and may be reinforced by including currently unrepresented processes, such as the stimulation of nitrogen fixation by increasing pCO₂⁶ and anthropogenic iron deposition⁵⁰. Net accumulation of bioavailable nitrogen in the 21st century, if realised, ...”*

Supplementary Note I: I think there are some errors in signs the fractionation factors.

Our factors largely follow common values in the literature (Sigman and Fripiat 2019), but we agree that their representation as negative rather than positive was against standard notation and have corrected this in our supplementary materials.

Altabet, M. A. (2007). Constraints on oceanic N balance/imbalance from sedimentary 15N records. *Biogeosciences* 4, 75–86. doi: 10.5194/bg-4-75-2007.

Krishnamurthy, A., J. K. Moore, N. Mahowald, C. Luo, and C. S. Zender (2010), Impacts of atmospheric nutrient inputs on marine biogeochemistry, *J. Geophys. Res.*, 115, G01006, doi:10.1029/2009JG001115.

Landolfi, A., C. J. Somes, W. Koeve, L. M. Zamora, and A. Oschlies (2017), Oceanic nitrogen cycling and N₂O flux perturbations in the Anthropocene, *Global Biogeochem. Cycles*, 31, 1236–1255, doi:10.1002/2017GB005633.

Moore, J. K., K. Lindsay, S. C. Doney, M. C. Long, and K. Misumi (2013), Marine ecosystem dynamics and biogeochemical cycling in the community Earth system model [CESM1(BGC)]: Comparison of the 1990s with the 2090s under the RCP4.5 and RCP8.5 scenarios, *J. Clim.*, 26(23), 9291–9312, doi:10.1175/JCLI-D-12-00566.1.

Somes, C. J., A. Landolfi, W. Koeve, and A. Oschlies (2016), Limited impact of atmospheric nitrogen deposition on marine productivity due to biogeochemical feedbacks in a global ocean model, *Geophys. Res. Lett.*, 43, 4500–4509, doi:10.1002/2016GL068335.

Yang, S., and N. Gruber (2016), The anthropogenic perturbation of the marine nitrogen cycle by atmospheric deposition: Nitrogen cycle feedbacks and the 15N Haber-Bosch effect, *Global Biogeochem. Cycles*, 30, 1418–1440, doi:10.1002/2016GB005421.

Battaglia, G., and F. Joos. 2018. Marine N₂O Emissions From Nitrification and Denitrification Constrained by Modern Observations and Projected in Multimillennial Global Warming Simulations. *Global Biogeochemical Cycles* 32: 92–121. doi:10.1002/2017GB005671.

Bindoff, N. L., W. W. L. Cheung, J. G. Kairo, J. Aristegui, V. A. Guinder, R. Hallberg, N. Hilmi, N. Jiao, et al. 2019. Changing Ocean, Marine Ecosystems, and Dependent Communities. In *IPCC Special*

Report on the Ocean and Cryosphere in a Changing Climate, ed. H.-O. Portner, C. D. Roberts, V. Masson-Delmotte, P. Zhai, E. Tignor, E. Poloczanska, K. Mintenbeck, A. Alegria, et al., 447–588. doi:<https://www.ipcc.ch/report/srocc/>.

- Deutsch, C., W. Berelson, R. Thunell, T. Weber, C. Tems, J. McManus, J. Crusius, T. Ito, et al. 2014. Centennial changes in North Pacific anoxia linked to tropical trade winds. *Science* 345: 665–668. doi:10.1126/science.1252332.
- Hamilton, D. S., R. A. Scanza, S. D. Rathod, T. C. Bond, J. F. Kok, L. Li, H. Matsui, and N. M. Mahowald. 2020. Recent (1980 to 2015) Trends and Variability in Daily-to-Interannual Soluble Iron Deposition from Dust, Fire, and Anthropogenic Sources. *Geophysical Research Letters* 47. doi:10.1029/2020GL089688.
- Kalvelage, T., G. Lavik, P. Lam, S. Contreras, L. Arteaga, C. R. Löscher, A. Oschlies, A. Paulmier, et al. 2013. Nitrogen cycling driven by organic matter export in the South Pacific oxygen minimum zone. *Nature Geoscience* 6. Nature Publishing Group: 228–234. doi:10.1038/ngeo1739.
- Kast, E. R., D. A. Stolper, A. Auderset, J. A. Higgins, H. Ren, X. T. Wang, A. Martinez-Garcia, G. H. Haug, et al. 2019. Nitrogen isotope evidence for expanded ocean suboxia in the early Cenozoic. *Science* 364: 386–389. doi:10.1126/science.aau5784.
- Krishnamurthy, A., J. K. Moore, N. Mahowald, C. Luo, S. C. Doney, K. Lindsay, and C. S. Zender. 2009. Impacts of increasing anthropogenic soluble iron and nitrogen deposition on ocean biogeochemistry. *Global Biogeochemical Cycles* 23: n/a-n/a. doi:10.1029/2008GB003440.
- Kwiatkowski, L., O. Aumont, and L. Bopp. 2019. Consistent trophic amplification of marine biomass declines under climate change. *Global Change Biology* 25: 218–229. doi:10.1111/gcb.14468.
- Kwiatkowski, L., O. Torres, L. Bopp, O. Aumont, M. Chamberlain, J. R. Christian, J. P. Dunne, M. Gehlen, et al. 2020. Twenty-first century ocean warming, acidification, deoxygenation, and upper-ocean nutrient and primary production decline from CMIP6 model projections. *Biogeosciences* 17: 3439–3470. doi:10.5194/bg-17-3439-2020.
- Landolfi, A., C. J. Somes, W. Koeve, L. M. Zamora, and A. Oschlies. 2017. Oceanic nitrogen cycling and N₂O flux perturbations in the Anthropocene. *Global Biogeochemical Cycles* 31: 1236–1255. doi:10.1002/2017GB005633.
- Letscher, R. T., F. Primeau, and J. K. Moore. 2016. Nutrient budgets in the subtropical ocean gyres dominated by lateral transport. *Nature Geoscience* 9. doi:10.1038/NGEO2812.
- Levy, M., L. Resplandy, J. B. Palter, D. Couespel, and Z. Lachkar. 2021. The crucial contribution of mixing to present and future ocean oxygen distribution. In *Ocean Mixing*, ed. A. C. Naveira-Garabato and M. P. Meredith.
- Moore, C. M., M. M. Mills, K. R. Arrigo, I. Berman-Frank, L. Bopp, P. W. Boyd, E. D. Galbraith, R. J. Geider, et al. 2013. Processes and patterns of oceanic nutrient limitation. *Nature Geoscience* 6. Nature Publishing Group: 701–710. doi:10.1038/ngeo1765.
- Moore, J. K., K. Lindsay, S. C. Doney, M. C. Long, and K. Misumi. 2013. Marine Ecosystem Dynamics and Biogeochemical Cycling in the Community Earth System Model [CESM1(BGC)]: Comparison of the 1990s with the 2090s under the RCP4.5 and RCP8.5 Scenarios. *Journal of Climate* 26: 9291–9312. doi:10.1175/JCLI-D-12-00566.1.
- Needoba, J. A., and P. J. Harrison. 2004. Influence of low light and a light:dark cycle on NO₃ uptake, intracellular NO₃, and nitrogen isotope fractionation by marine phytoplankton. *Journal of Phycology* 40: 505–516. doi:10.1111/j.1529-8817.2004.03171.x.

- Rafter, P. A., A. Bagnell, D. Marconi, and T. DeVries. 2019. Global trends in marine nitrate N isotopes from observations and a neural network-based climatology. *Biogeosciences* 16: 2617–2633. doi:10.5194/bg-16-2617-2019.
- Ren, H., D. M. Sigman, R. C. Thunell, and M. G. Prokopenko. 2012. Nitrogen isotopic composition of planktonic foraminifera from the modern ocean and recent sediments. *Limnology and Oceanography* 57: 1011–1024. doi:10.4319/lo.2012.57.4.1011.
- Ren, H., Y.-C. Chen, X. T. Wang, G. T. F. Wong, A. L. Cohen, T. M. DeCarlo, M. A. Weigand, H.-S. Mii, et al. 2017. 21st-century rise in anthropogenic nitrogen deposition on a remote coral reef. *Science* 356: 749–752. doi:10.1126/science.aal3869.
- Sigman, D. M., and F. Fripiat. 2019. Nitrogen Isotopes in the Ocean. In *Encyclopedia of Ocean Sciences*, 263–278. Elsevier. doi:10.1016/B978-0-12-409548-9.11605-7.
- Sloyan, B. M., and I. V. Kamenkovich. 2007. Simulation of Subantarctic Mode and Antarctic Intermediate Waters in Climate Models. *Journal of Climate* 20: 5061–5080. doi:10.1175/JCLI4295.1.
- Somes, C. J., A. Schmittner, E. D. Galbraith, M. F. Lehmann, M. A. Altabet, J. P. Montoya, R. M. Letelier, A. C. Mix, et al. 2010. Simulating the global distribution of nitrogen isotopes in the ocean. *Global Biogeochemical Cycles* 24: 1–16. doi:10.1029/2009GB003767.
- Somes, C. J., A. Landolfi, W. Koeve, and A. Oschlies. 2016. Limited impact of atmospheric nitrogen deposition on marine productivity due to biogeochemical feedbacks in a global ocean model. *Geophysical Research Letters* 43: 4500–4509. doi:10.1002/2016GL068335.
- Straub, M., D. M. Sigman, H. Ren, A. Martínez-García, A. N. Meckler, M. P. Hain, and G. H. Haug. 2013. Changes in North Atlantic nitrogen fixation controlled by ocean circulation. *Nature* 501: 200–203. doi:10.1038/nature12397.
- Wannicke, N., C. Frey, C. S. Law, and M. Voss. 2018. The response of the marine nitrogen cycle to ocean acidification. *Global Change Biology* 24: 5031–5043. doi:10.1111/gcb.14424.
- Yang, S., and N. Gruber. 2016. The anthropogenic perturbation of the marine nitrogen cycle by atmospheric deposition: Nitrogen cycle feedbacks and the ^{15}N Haber-Bosch effect. *Global Biogeochemical Cycles* 30: 1418–1440. doi:10.1002/2016GB005421.

Reviewer 2

This manuscript by Buchanan et al. uses the PISCES global ocean biogeochemical model with nitrogen isotopes to project changes of atmospheric nitrogen deposition and climate on changes to the $\delta^{15}\text{N}$ distribution. Their main finding is that reduced productivity driven by climate change effects is detectable through significantly decreased twilight zone $\delta^{15}\text{NO}_3$. Their simulations also predict that climate change is increasing the N budget.

I think this is a really nice set of model experiments that a lot can be learned from and important conclusions can be made, making it a nice study for this journal. The manuscript is well written and figures clearly presented. However, I find the main interpretation of the model results somewhat unconvincing and explanation oversimplified (see comments below). On the other hand, I think the Time of Emergence analysis makes a very important point about the usefulness of $\delta^{15}\text{N}$ which is not typically included in models. I think after some clarifications and revisions this can be an excellent manuscript for publication.

Best regards,
Christopher Somes
GEOMAR Helmholtz Centre for Ocean Research Kiel

We want to thank Chris Somes for his positive and useful comments. Like the other reviewers, Dr Somes has asked for a clearer explanation of the mechanism causing the $\delta^{15}\text{N}$ declines and have asked us to acknowledge prior studies on this topic. We address their comments in our responses below.

Major Comments:

lines 58-62: "In general, NPP and sinks increase $\delta^{15}\text{N}$ "
"... nitrogen isotopes to probe past changes in nitrogen cycling and NPP"

Very rarely is a direct link made with $\delta^{15}\text{N}$ and NPP, it is typically $\delta^{15}\text{N}$ and surface DIN utilization (in the absence of source/sink processes), which is a mechanism that operates much differently than NPP. For example, some of the highest $\delta^{15}\text{N}$ values occur in the low productivity gyres (as long as N_2 fixation is not occurring) due to high DIN utilization from depletion, while there is a notable minimum in $\delta^{15}\text{N}$ across the highly productive equatorial Pacific due to low utilization caused by iron limitation (e.g. [Altabet and Francois, 1994]).

The paleo studies cited (22-24) all focus on external sources N_2 fixation and denitrification, which are affected by and interact with NPP but are far from a direct proxy of NPP. Even considering a region far removed from external source/sink processes such as the glacial Southern Ocean is quite complicated, where productivity proxies show higher productivity in the Subantarctic zones versus lower productivity in the higher latitude Antarctic zone (e.g. [Kohfeld et al., 2005]). However, the $\delta^{15}\text{N}$ signal shows a fairly consistent increase throughout both zones, which can be explained by higher utilization due to iron deposition in the SAZ and lower supply via enhanced stratification in the AZ ([Kemeny et al., 2018] [François et al., 1997; Robinson and Sigman, 2008]). I bring up this example because I worry that non-experts may think a simple relationship between NPP and $\delta^{15}\text{N}$ exists, which is not always the case and I think the highly variable nature of the surface $\delta^{15}\text{N}$ model results support this as well.

We agree with the reviewer. These changes in language have been made in the title, abstract, introduction and throughout the manuscript. Additional clarity in this respect is also found from our clearer explanation of the mechanism in Section 3, which has now been split into two sections (3 and 4):

Section 3 (Line 161): “Linking nitrogen cycling and isotopic signals”

Section 4 (Line 229): “Warming versus circulation changes”

lines 161-163: “growing nitrogen limitation across the lower latitude caused more ^{15}N to be removed from the euphotic zone in sinking organic matter and therefore depleted ^{15}N within the remaining dissolved bioavailable nitrogen.”

I’m having trouble understanding how this mechanism operates. When view spatial gradients, typically the opposite happens. Phytoplankton preferentially incorporate the ^{14}N when it is readily available, leaving ^{15}N -enriched DIN to advect into the gyres. For example, this is why you see a minimum of $\delta^{15}\text{N}$ -PON in the productive equatorial Pacific that trends to higher values in the more N-limited gyres ([Altabet and Francois, 1994]). I’m confused why this is not occurring in the model.

We agree that the mechanism could be more clearly explained. The revised manuscript has improved both the schematic of Figure 3d by adding numbers that are used in the text and caption for a clearer explanation and has edited and split section 3 into two sections. Both revisions hopefully have improved the clarity of the mechanisms at play.

Section 3 (lines 161 – 227) now reads:

“While low $\delta^{15}\text{N}$ from nitrogen deposition was important for 20th century declines in $\delta^{15}\text{N}_{\text{NO}_3}$, consistent with previous modelling³⁴, climate-driven declines in the 21st century potentially involved changes to numerous nitrogen cycle processes, the individual effects of which are difficult to isolate. Potential contributors included a decrease in phytoplankton production and fractionation due to nitrogen limitation, an increase in nitrogen fixation, an increase in zooplankton recycling, a decrease in denitrification, or a physical redistribution of the dissolved nitrogen compounds by a changing circulation (or any combination of these processes). Here, we demonstrate that changes ultimately linked to increasing nitrogen limitation, namely a decrease in phytoplankton production and fractionation potentially supplemented by a tropical-subtropical shift of nitrogen fixation, was the primary driver of the broad, simulated $\delta^{15}\text{N}_{\text{NO}_3}$ declines that emerged in the twilight zones of the low latitude oceans.

Firstly, we infer that ^{15}N -depleted organic matter in the euphotic zone drove the $\delta^{15}\text{N}_{\text{NO}_3}$ declines in the twilight zone. Widespread declines in $\delta^{15}\text{N}_{\text{POM}}$ (Figure 3a) delivered less ^{15}N to the twilight zone via the remineralisation of sinking organic material. This inference was further supported by an analysis of purely biogeochemical $^{15}\text{NO}_3$ fluxes in the twilight zone, which disentangled local biogeochemical effects from circulation effects (see Methods), and was able to reproduce the depletion of $\delta^{15}\text{N}_{\text{NO}_3}$ across the tropics and subtropics (Figure 3b). An exception were regions influenced by Southern Ocean mode waters in the South Pacific and Indian oceans⁴³ where upstream increases in NPP and nitrogen consumption in response to climate change (Figure 1c) led to a biogeochemically driven enrichment in $\delta^{15}\text{N}_{\text{NO}_3}$. However, the broad biogeochemical tendency was to

lower twilight zone $\delta^{15}\text{N}_{\text{NO}_3}$ and this was linked to depleted $\delta^{15}\text{N}_{\text{POM}}$ sinking out of subtropical euphotic zones.

Secondly, the decline in nitrogen assimilation rates by phytoplankton (by 297 Tg N yr⁻¹ globally; Figure 1c) far exceeded parallel changes in the rate of sources and sinks, and made an important contribution to the simulated isotopic declines. Increasing nitrogen limitation of phytoplankton was responsible for reduced nitrogen assimilation across the lower latitude oceans during the 21st century in our model, with bioavailable nitrogen declining by 0.5-4 mmol m⁻³ (5-40%) in the tropical Pacific and Atlantic euphotic zones (by 2081-2100 relative to 1986-2005; Figure 3c). These trends are consistent with the common projections of declining upper ocean nitrate inventories across ESM studies that are largest in the low latitude upwelling systems^{10,11}. This is important because changing nitrogen availability not only affects the success of nitrogen fixers, but also affects the rate of nitrogen uptake by phytoplankton and the degree of nitrogen isotope fractionation, which combine to control the horizontal $\delta^{15}\text{N}_{\text{NO}_3}$ gradients across the low latitude ocean^{1,20,22,23}. Increasing nitrogen limitation under climate change therefore led to both a stimulation of nitrogen fixation and a weaker isotopic enrichment associated with phytoplankton assimilation. Weaker fractionation by phytoplankton (i.e. a weaker preference for ¹⁴N) meant that more ¹⁵N was used to create organic matter in increasingly nitrogen-limited upwelling regions, and was subsequently lost via sinking (step 1 in Figure 3d). Consequently, euphotic zone $\delta^{15}\text{N}_{\text{NO}_3}$ and $\delta^{15}\text{N}_{\text{POM}}$ values declined at the boundary between the nitrogen-replete and nitrogen-deplete regimes (step 2 in Figure 3d). Because subtropical gyres receive most of their nutrients from lateral transport²⁴, ¹⁵N-depleted nitrogen was then swept into the subtropical gyres where local nitrogen recycling and organic matter formation proceeded with lighter isotopic signatures (i.e. relatively more ¹⁴N; step 3 in Figure 3d), and, combined with local increases in biological nitrogen fixation, delivered lower $\delta^{15}\text{N}_{\text{POM}}$ values to the subtropical twilight zone.

The removal of ¹⁵N by nitrogen limited phytoplankton in the tropics followed by the transfer of ¹⁵N-depleted water to the gyres is also supported by idealised modelling. We constructed a zero-dimensional (0D) model that follows nitrogen uptake and fractionation by phytoplankton in a water parcel (see Methods; Supplementary Table 1). In this simple model, an inorganic nitrogen pool representing initial upwelled nitrogen is steadily assimilated by phytoplankton to create organic matter, and this organic matter is either remineralised back to inorganic nitrogen or removed permanently via export. For every 10% decline in initial upwelled nitrogen supplied to the water parcel (corresponding to a ~10% loss in integrated NPP during the water parcel's lifetime), there is a $0.19 \pm 0.03\%$ decline in the final $\delta^{15}\text{N}_{\text{POM}}$ produced within the gyre once nitrogen is depleted to limiting concentrations. The uncertainty of $\pm 0.03\%$ is associated with temperature changes of $\pm 4^\circ\text{C}$, suggesting that warming also plays a role by modulating growth rates and recycling (Supplementary Figure 9). Extrapolating this 'rule of thumb' to the global model, we expect $\delta^{15}\text{N}_{\text{POM}}$ declines of 0.1-0.8‰ for bioavailable nitrogen declines of 5-40%, which agrees broadly with the results of the full model by 2081-2100 (Figure 2c). Furthermore, the local enrichment of $\delta^{15}\text{N}_{\text{POM}}$ in the upwelling region in both our 0D model (compare solid and dashed lines in Figure 3d) and in the tropical Pacific in our global model (Figure 3a) clearly signifies the existence of this mechanism, where more ¹⁵N was removed from upwelling zones due to nitrogen limitation of phytoplankton (note that this feature was absent in the Atlantic due to local declines in denitrification (Figure 1g)). Ultimately, the biogeochemical consequences of increasing nitrogen limitation appeared to be the primary cause of the widespread isotopic declines."

Section 4 (lines 229 – 247) now reads:

“We examined the direct (i.e. warming on biogeochemical rates) and indirect effects (i.e. circulation altering nutrient supply) of climate change in two additional experiments. First, warming was imposed on biogeochemical processes under otherwise preindustrial conditions to mimic its effect on rates. Second, preindustrial temperatures were imposed while climate change altered the circulation to mimic its effect on substrate availability (see Methods). Direct effects of warming on biogeochemical processes showed a limited ability to reproduce the full suite of climate-driven trends, with the exception of the poleward shift in nitrogen fixers (Figure 4). In contrast, changes to ocean circulation (e.g. stratification) in the indirect effect simulation replicated changes in euphotic zone nitrate (Spearman’s rank correlation; $r_s = 0.99$), twilight zone $\delta^{15}\text{NNO}_3$ ($r_s = 0.94$), twilight zone $\delta^{15}\text{NPOM}$ ($r_s = 0.95$), NPP ($r_s = 0.69$), zooplankton grazing ($r_s = 0.69$) and water column denitrification ($r_s = 0.88$) and sedimentary denitrification ($r_s = 0.90$) seen in the full model (Figure 4). Moderate agreement for nitrogen fixation in both the direct effect ($r_s = 0.65$) and indirect effect simulations ($r_s = 0.61$) suggested that both warming and circulation changes were equally important in determining its full response, but that nitrogen fixation by itself was not sufficient to force the isotopic trends associated with climate change. The effect of climate change on biogeochemical processes via altering substrate availability was therefore a major cause of 21st century increases in nitrogen limitation, resultant NPP declines, shifts in the sources and sinks of nitrogen, and the accompanying trends in nitrogen isotopes that fingerprint the response of the nitrogen cycle.”

New schematic (change is the numbers for aid in explanation):

In supplementary Figure 8 (OD model), what is the delta d15N-POM reflecting? Is it total POM accumulated throughout the entire “transport” to the DIN depleted gyre or the final d15N-POM in the gyre? It is confusing since the x-axis is initial bioavailable DIN and not decreasing DIN during transport to the gyre - I’m not sure what the d15N-POM signature is when DIN is nearly fully depleted. If DIN is eventually fully depleted, the total d15N-POM produced should equal the original supply due to mass balance so I’m not sure how this continuous d15N decrease can occur.

The $\delta^{15}\text{N}_{\text{POM}}$ in Supplementary Figure 8 (now Supp Figure 9) is reflecting the final equilibrium value of $\delta^{15}\text{N}_{\text{POM}}$ given the initial N concentration. Confusion over what this figure is showing was also expressed by reviewer 1, and we have therefore made the caption clearer.

Also, our OD model is constructed to conserve mass.

I guess what may be happening is when you start with less DIN, the amount of DIN consumed during depletion towards to gyre is less, which reduces utilization and thus also reduces enrichment of d15N causing lower delta d15N in the different scenarios with lower initial DIN (yes!). But the

spatial/temporal d15N trend from the upwelling zone to the gyre has to increase in each individual scenario as DIN is depleted because phytoplankton still first preferentially consume 14N, right? I'm not sure I'm understanding it correctly, I think it would be helpful to show the transient changes of d15N-POM as the system is becoming more DIN depleted from the upwelling zone towards the gyre, not only the steady-state results based on initial DIN.

We hope that the revised schematic and explanation make this clearer, as addressed above.

I also wonder if applying an open system fractionation system to the OD experiment is realistic. You noted there is loss due to sinking POM so the system is not completely closed, but the DIN source is cut-off so I think the closed system fractionation would make more sense and ensure total 15N mass conservation from d15N-DIN to d15N-POM.

An open fractionation system is essential because it allows particulate matter to escape the hypothetical water parcel, the main process that removes nutrients from surface waters in the lower latitudes. Mass is conserved, as we track the amount of ^{14}N and ^{15}N that is removed.

I'm skeptical about to what degree the NPP mechanism in Figure 3d is actually occurring in the global model given the highly variable surface d15N signal (Figure 3a, Supp. Fig. 7a). And looking more closely, the largest area of decreased NPP is the central equatorial Pacific (Figure 1c), but there is hardly any decrease in twilight zone d15NO₃ there (Figure 2a). The largest decrease of twilight zone d15NO₃ occurs in the subtropical western/central North Pacific, but at the surface there is very little change and even a slight increase in NPP in the northern part of this signal. It seems to me there is a lot more going on than a simple relationship to NPP.

We hope the new schematic and edits makes this mechanism clearer. We acknowledge that other contributions may be occurring, and we have modified the manuscript to acknowledge them. As a brief additional point, we would point the reviewer towards Figure 3a, where the enrichment of d15N_{POM} occurs in the eastern Pacific near the upwelling zone. This feature is consistent with our proposed mechanism, which we now explicitly talk to in section 3.

N₂ fixation and N deposition effects

I see a much stronger spatial match of the combined increases of N₂ fixation and N deposition with the decreased twilight zone delta d15NO₃. I think you have a complicated surface utilization signal, which is completely expected due to all the processes that control utilization and how they all differ regionally. Then deeper in the twilight zone, you are beginning see this noted net nitrogen accumulation of low d15N near the source increases since they introduce very low d15N into the system. Of course it is difficult for me to understand everything here, but I am surprised these processes are not considered to be significantly contributing the decreased d15NO₃, although lines 210-211 briefly mention this potential effect from N₂ fixation. Have you checked if these areas of decreased twilight zone d15NO₃ are associated with nitrogen accumulation?

My past modeling has demonstrated that even small changes in N₂ fixation in the N deprived gyres can have large impacts on d15N since there is little to begin with, e.g. shown by modifying their iron limitation in the modern [Somes et al., 2010] and LGM [Somes et al., 2017](please don't feel obligated to cite these, I mention them to support my statements on the importance of N₂ fixation), and I think N deposition would have a similar importance where rates are high.

The reviewer is correct that there are alternative explanations, such as denitrification, N₂ fixation, zooplankton recycling and the redistribution of isotopes by the circulation. We therefore agree that these alternatives must be addressed, and we have addressed them in the revised section 3.

Lines 165 – 168: *“Potential contributors included a decrease in phytoplankton production and fractionation due to nitrogen limitation, an increase in nitrogen fixation, an increase in zooplankton recycling, a decrease in denitrification, or a physical redistribution of the dissolved nitrogen compounds by a changing circulation (or any combination of these processes).”*

With regard to the N accumulation question: We show below the lack of correlation between changes in twilight d₁₅N_{NO₃} and NO₃ concentrations. Therefore, whether NO₃ accumulates or decreases in a region of the low latitude twilight zone appears to have little relationship with the d₁₅N_{NO₃} trends.

Time of Emergence

As mentioned above, I think this is a fantastic point to make that will be very important to the broader marine biogeochemistry community since it is often so hard to quantify trends with noisy and sparse rate measurements.

We thank Dr Somes for his positive comment.

Other Literature

Please replace the [Somes et al., 2016] citation with [Landolfi et al., 2017], which is much more relevant here since the latter study also includes warming and N deposition individually and combined, very similar to here (albeit without isotopes). I think it would be useful to have one paragraph noting key differences with the Landolfi et al (2017) and Yang and Gruber (2016) studies in the main text, but I leave this decision up to you since space may be tight.

We agree that the Landolfi et al. (2017) paper is very relevant and have expanded our discussion in the first section (“anthropogenic alteration”) and the discussion to acknowledge this study.

We also cite the Yang and Gruber (2016) paper.

For example, the negative feedback of N fixation in response to N deposition is much weaker in your model, allowing increased N₂ fixation and more net nitrogen accumulation to occur in the climate+N deposition scenario compared to Landolfi et al (2017), although our model did predict a slightly higher N inventory in the combined simulation. I would be interested more generally on what is driving this increase in N₂ fixation, but perhaps that would take up too much space since the focus is

on N isotopes, so again this is your decision. The Yang and Gruber (2016) study seems to show a stronger effect from N deposition alone on $\delta^{15}\text{N}$ than your model (at least that is the impression I get from reading their paper compared to yours), I think it would also be useful to comment on this.

We agree with the reviewer and have acknowledged the Landolfi et al. (2017) study in section 1 “Anthropogenic alteration” and in the discussion.

With regard to the Yang and Gruber (2016) study: they found a substantially greater effect on $\delta^{15}\text{N}_{\text{NO}_3}$ because their $\delta^{15}\text{N}$ signatures of NO_x and NH_y were very low (-7‰ for NO_x and -10‰ for NH_y in their base case scenario). These values are towards the lower end of $\delta^{15}\text{N}$ measurements of inorganic and organic aerosol N which range from -15 to $+10 \text{‰}$ and -5 to $+15\text{‰}$ (Sigman and Fripiat, 2019).

Figures

The figures are well presented. But I think one reason contributing to my difficulty to understand this proposed mechanism is every single figure shows $\Delta \delta^{15}\text{N}$. Especially since this is the first use of nitrogen isotopes in the PISCES model, I think the initial preindustrial spatial $\delta^{15}\text{N}$ distributions that the projected changes arise from should also be shown in the supplementary material, not only statistical metrics.

We agree, Figure 1 in the supplementary information now shows the global maps of $\delta^{15}\text{N}_{\text{NO}_3}$ and $\delta^{15}\text{N}_{\text{POM}}$ averaged in the euphotic zone and averaged between 1986-2005 under the full anthropogenic scenario (Climate change + N deposition).

Concluding Remarks

I think this is a really interesting study, especially with the important point of the Time of Emergence. However it is still important to get the mechanism(s) correct. Even if I am missing and/or not understanding something correctly and you are convinced about your general interpretation/mechanism, I still think it should be mainly explained in terms of changes to surface DIN utilization, rather than directly relating it to NPP. And to be fair, fractionation and N limitation are briefly mentioned in the explanation, but the focus on NPP in the introduction and conclusion will give too much of an oversimplified view in my opinion to the more general scientific audience.

We agree with Chris and have revised the manuscript accordingly. We are also confident in our proposed mechanism based on our arguments and mechanisms provided above.

Altabet, M. A., and R. Francois (1994), Sedimentary nitrogen isotopic ratio as a recorder for surface ocean nitrate utilization, *Global Biogeochem. Cycles*, 8(1), 103-116, doi: 10.1029/93gb03396.

François, R., M. A. Altabet, E.-F. Yu, D. M. Sigman, M. P. Bacon, M. Frank, G. Bohrmann, G. Bareille, and L. D. Labeyrie (1997), Contribution of Southern Ocean surface-water stratification to low atmospheric CO_2 concentrations during the last glacial period, *Nature*, 389(6654), 929-935, doi: 10.1038/40073.

Kemeny, P. C., E. R. Kast, M. P. Hain, S. E. Fawcett, F. Fripiat, A. S. Studer, A. Martinez-Garcia, G. H. Haug, and D. M. Sigman (2018), A Seasonal Model of Nitrogen Isotopes in the Ice Age Antarctic Zone: Support for Weakening of the Southern Ocean Upper Overturning Cell, *Paleoceanography and*

Paleoclimatology, 33(12), 1453-1471, doi: 10.1029/2018pa003478.

Kohfeld, K. E., C. Le Quere, S. P. Harrison, and R. F. Anderson (2005), Role of marine biology in glacial-interglacial CO₂ cycles, *Science*, 308(5718), 74-78, doi: 10.1126/science.1105375.

Landolfi, A., C. J. Somes, W. Koeve, L. M. Zamora, and A. Oschlies (2017), Oceanic nitrogen cycling and N₂O flux perturbations in the Anthropocene, *Global Biogeochemical Cycles*, 31(8), 1236-1255, doi: 10.1002/2017gb005633.

Robinson, R. S., and D. M. Sigman (2008), Nitrogen isotopic evidence for a poleward decrease in surface nitrate within the ice age Antarctic, *Quaternary Science Reviews*, 27(9-10), 1076-1090, doi: 10.1016/j.quascirev.2008.02.005.

Somes, C. J., A. Schmittner, and M. A. Altabet (2010), Nitrogen isotope simulations show the importance of atmospheric iron deposition for nitrogen fixation across the Pacific Ocean, *Geophys. Res. Lett.*, 37(23), L23605, doi: 10.1029/2010gl044537.

Somes, C. J., A. Landolfi, W. Koeve, and A. Oschlies (2016), Limited impact of atmospheric nitrogen deposition on marine productivity due to biogeochemical feedbacks in a global ocean model, *Geophysical Research Letters*, 43, 4500-4509, doi: 10.1002/2016gl068335.

Somes, C. J., A. Schmittner, J. Muglia, and A. Oschlies (2017), A three-dimensional model of the marine nitrogen cycle during the Last Glacial Maximum constrained by sedimentary isotopes, *Frontiers in Marine Science*, 4(108), doi: 10.3389/fmars.2017.00108.

Bindoff, N. L., Cheung, W. W. L., Kairo, J. G., Aristegui, J., Guinder, V. A., Hallberg, R., et al. (2019). "Changing Ocean, Marine Ecosystems, and Dependent Communities," in *IPCC Special Report on the Ocean and Cryosphere in a Changing Climate*, eds. H.-O. Portner, C. D. Roberts, V. Masson-Delmotte, P. Zhai, E. Tignor, E. Poloczanska, et al., 447–588. doi:<https://www.ipcc.ch/report/srocc/>.

Kwiatkowski, L., Torres, O., Bopp, L., Aumont, O., Chamberlain, M., Christian, J. R., et al. (2020). Twenty-first century ocean warming, acidification, deoxygenation, and upper-ocean nutrient and primary production decline from CMIP6 model projections. *Biogeosciences* 17, 3439–3470. doi:10.5194/bg-17-3439-2020.

Letscher, R. T., Primeau, F., and Moore, J. K. (2016). Nutrient budgets in the subtropical ocean gyres dominated by lateral transport. *Nat. Geosci.* 9. doi:10.1038/NGEO2812.

Moore, C. M., Mills, M. M., Arrigo, K. R., Berman-Frank, I., Bopp, L., Boyd, P. W., et al. (2013). Processes and patterns of oceanic nutrient limitation. *Nat. Geosci.* 6, 701–710. doi:10.1038/ngeo1765.

Needoba, J. A., and Harrison, P. J. (2004). Influence of low light and a light:dark cycle on NO₃ uptake, intracellular NO₃, and nitrogen isotope fractionation by marine phytoplankton. *J. Phycol.* 40, 505–516. doi:10.1111/j.1529-8817.2004.03171.x.

Rafter, P. A., Bagnell, A., Marconi, D., and DeVries, T. (2019). Global trends in marine nitrate N isotopes from observations and a neural network-based climatology. *Biogeosciences* 16, 2617–2633. doi:10.5194/bg-16-2617-2019.

- Sigman, D. M., and Fripiat, F. (2019). "Nitrogen Isotopes in the Ocean," in *Encyclopedia of Ocean Sciences* (Elsevier), 263–278. doi:10.1016/B978-0-12-409548-9.11605-7.
- Sloyan, B. M., and Kamenkovich, I. V. (2007). Simulation of Subantarctic Mode and Antarctic Intermediate Waters in Climate Models. *J. Clim.* 20, 5061–5080. doi:10.1175/JCLI4295.1.
- Somes, C. J., Schmittner, A., Galbraith, E. D., Lehmann, M. F., Altabet, M. A., Montoya, J. P., et al. (2010). Simulating the global distribution of nitrogen isotopes in the ocean. *Global Biogeochem. Cycles* 24, 1–16. doi:10.1029/2009GB003767.
- Yang, S., and Gruber, N. (2016). The anthropogenic perturbation of the marine nitrogen cycle by atmospheric deposition: Nitrogen cycle feedbacks and the ^{15}N Haber-Bosch effect. *Global Biogeochem. Cycles* 30, 1418–1440. doi:10.1002/2016GB005421.

Reviewer 3

In this study, the authors model the impact of climate change and anthropogenically-driven increases in atmospheric nitrogen deposition on net primary production and the accumulation of fixed nitrogen in the ocean. Surprisingly (to me), they found that the RCP8.5 climate change scenario led to a larger change in the nitrogen inventory than the nitrogen deposition flux. Specifically, climate change led to an accumulation of nitrogen in the ocean through enhanced stratification, decreasing nutrient delivery to the euphotic zone and driving a decrease in NPP. The authors use a novel approach based on N isotope patterns in nitrate and particulate organic matter to address an important question about the future evolution of the marine N cycle. I have some questions for further clarification of their methods, results, and discussion.

We want to thank the reviewer for their positive and useful comments. They, like the other reviewers, have asked for a clearer explanation of the mechanism causing the $\delta^{15}\text{N}$ declines. We address their comments in our responses below.

Lines 67-70: I thought it would be helpful if these references to 'climate change' included some of the more specific drivers of the phenomenon being assessed. It's difficult to compare here the impact of a complex suite of environmental changes due to 'climate change' with a very specific quantifiable process such as anthropogenic increases in N deposition.

We have expanded on how climate change affects the N cycle earlier in the introduction:

Lines 50 - 52: *"For instance, key nitrogen cycle processes will be altered directly as warming changes biogeochemical rates (i.e. metabolism)^{17,18}, and indirectly as circulation alters substrate supply¹⁹."*

Lines 72 – 75: *"We quantify the relative roles of climate change under the high emissions RCP8.5 scenario²⁹ and anthropogenic increases in nitrogen deposition⁸, assess the ToE of nitrogen isotopes, and disentangle the direct (warming on biogeochemical rates) and indirect (substrate supply on biogeochemical rates) effects of climate change to reveal the primary drivers of change."*

Lines 157-165: This section was a little counter-intuitive to me and could probably be clarified. Especially with line 162, where increasing N utilization should lead to more export of ^{15}N , and a lower concentration of ^{15}N in the euphotic zone, but I would expect a higher $^{15}\text{N}/^{14}\text{N}$ ratio.

We agree that the mechanism could be more clearly explained. The revised manuscript has improved both the schematic of Figure 3d by adding numbers that are used in the text and caption for a clearer explanation and has edited section 3 and split it into two sections. Both revisions hopefully have improved the clarity of the mechanism as play.

Section 3 (lines 161 – 227) now reads:

“While low $\delta^{15}\text{N}$ from nitrogen deposition was important for 20th century declines in $\delta^{15}\text{N}_{\text{NO}_3}$, consistent with previous modelling³⁴, climate-driven declines in the 21st century potentially involved changes to numerous nitrogen cycle processes, the individual effects of which are difficult to isolate. Potential contributors included a decrease in phytoplankton production and fractionation due to nitrogen limitation, an increase in nitrogen fixation, an increase in zooplankton recycling, a decrease in denitrification, or a physical redistribution of the dissolved nitrogen compounds by a changing circulation (or any combination of these processes). Here, we demonstrate that changes ultimately linked to increasing nitrogen limitation, namely a decrease in phytoplankton production and fractionation potentially supplemented by a tropical-subtropical shift of nitrogen fixation, was the

primary driver of the broad, simulated $\delta^{15}\text{N}_{\text{NO}_3}$ declines that emerged in the twilight zones of the low latitude oceans.

Firstly, we infer that ^{15}N -depleted organic matter in the euphotic zone drove the $\delta^{15}\text{N}_{\text{NO}_3}$ declines in the twilight zone. Widespread declines in $\delta^{15}\text{N}_{\text{POM}}$ (Figure 3a) delivered less ^{15}N to the twilight zone via the remineralisation of sinking organic material. This inference was further supported by an analysis of purely biogeochemical $^{15}\text{NO}_3$ fluxes in the twilight zone, which disentangled local biogeochemical effects from circulation effects (see Methods), and was able to reproduce the depletion of $\delta^{15}\text{N}_{\text{NO}_3}$ across the tropics and subtropics (Figure 3b). An exception were regions influenced by Southern Ocean mode waters in the South Pacific and Indian oceans⁴³ where upstream increases in NPP and nitrogen consumption in response to climate change (Figure 1c) led to a biogeochemically driven enrichment in $\delta^{15}\text{N}_{\text{NO}_3}$. However, the broad biogeochemical tendency was to lower twilight zone $\delta^{15}\text{N}_{\text{NO}_3}$ and this was linked to depleted $\delta^{15}\text{N}_{\text{POM}}$ sinking out of subtropical euphotic zones.

Secondly, the decline in nitrogen assimilation rates by phytoplankton (by 297 Tg N yr⁻¹ globally; Figure 1c) far exceeded parallel changes in the rate of sources and sinks, and made an important contribution to the simulated isotopic declines. Increasing nitrogen limitation of phytoplankton was responsible for reduced nitrogen assimilation across the lower latitude oceans during the 21st century in our model, with bioavailable nitrogen declining by 0.5-4 mmol m⁻³ (5-40%) in the tropical Pacific and Atlantic euphotic zones (by 2081-2100 relative to 1986-2005; Figure 3c). These trends are consistent with the common projections of declining upper ocean nitrate inventories across ESM studies that are largest in the low latitude upwelling systems^{10,11}. This is important because changing nitrogen availability not only affects the success of nitrogen fixers, but also affects the rate of nitrogen uptake by phytoplankton and the degree of nitrogen isotope fractionation, which combine to control the horizontal $\delta^{15}\text{N}_{\text{NO}_3}$ gradients across the low latitude ocean^{1,20,22,23}. Increasing nitrogen limitation under climate change therefore led to both a stimulation of nitrogen fixation and a weaker isotopic enrichment associated with phytoplankton assimilation. Weaker fractionation by phytoplankton (i.e. a weaker preference for ^{14}N) meant that more ^{15}N was used to create organic matter in increasingly nitrogen-limited upwelling regions, and was subsequently lost via sinking (step 1 in Figure 3d). Consequently, euphotic zone $\delta^{15}\text{N}_{\text{NO}_3}$ and $\delta^{15}\text{N}_{\text{POM}}$ values declined at the boundary between the nitrogen-replete and nitrogen-deplete regimes (step 2 in Figure 3d). Because subtropical gyres receive most of their nutrients from lateral transport²⁴, ^{15}N -depleted nitrogen was then swept into the subtropical gyres where local nitrogen recycling and organic matter formation proceeded with lighter isotopic signatures (i.e. relatively more ^{14}N ; step 3 in Figure 3d), and, combined with local increases in biological nitrogen fixation, delivered lower $\delta^{15}\text{N}_{\text{POM}}$ values to the subtropical twilight zone.

The removal of ^{15}N by nitrogen limited phytoplankton in the tropics followed by the transfer of ^{15}N -depleted water to the gyres is also supported by idealised modelling. We constructed a zero-dimensional (0D) model that follows nitrogen uptake and fractionation by phytoplankton in a water parcel (see Methods; Supplementary Table 1). In this simple model, an inorganic nitrogen pool representing initial upwelled nitrogen is steadily assimilated by phytoplankton to create organic matter, and this organic matter is either remineralised back to inorganic nitrogen or removed permanently via export. For every 10% decline in initial upwelled nitrogen supplied to the water parcel (corresponding to a ~10% loss in integrated NPP during the water parcel's lifetime), there is a $0.19 \pm 0.03\text{‰}$ decline in the final $\delta^{15}\text{N}_{\text{POM}}$ produced within the gyre once nitrogen is depleted to limiting concentrations. The uncertainty of $\pm 0.03\text{‰}$ is associated with temperature changes of $\pm 4^\circ\text{C}$, suggesting that warming also plays a role by modulating growth rates and recycling (Supplementary

Figure 9). Extrapolating this ‘rule of thumb’ to the global model, we expect $\delta^{15}\text{N}_{\text{POM}}$ declines of 0.1-0.8‰ for bioavailable nitrogen declines of 5-40%, which agrees broadly with the results of the full model by 2081-2100 (Figure 2c). Furthermore, the local enrichment of $\delta^{15}\text{N}_{\text{POM}}$ in the upwelling region in both our OD model (compare solid and dashed lines in Figure 3d) and in the tropical Pacific in our global model (Figure 3a) clearly signifies the existence of this mechanism, where more ^{15}N was removed from upwelling zones due to nitrogen limitation of phytoplankton (note that this feature was absent in the Atlantic due to local declines in denitrification (Figure 1g)). Ultimately, the biogeochemical consequences of increasing nitrogen limitation appeared to be the primary cause of the widespread isotopic declines.”

Section 4 (lines 229 – 247) now reads:

“We examined the direct (i.e. warming on biogeochemical rates) and indirect effects (i.e. circulation altering nutrient supply) of climate change in two additional experiments. First, warming was imposed on biogeochemical processes under otherwise preindustrial conditions to mimic its effect on rates. Second, preindustrial temperatures were imposed while climate change altered the circulation to mimic its effect on substrate availability (see Methods). Direct effects of warming on biogeochemical processes showed a limited ability to reproduce the full suite of climate-driven trends, with the exception of the poleward shift in nitrogen fixers (Figure 4). In contrast, changes to ocean circulation (e.g. stratification) in the indirect effect simulation replicated changes in euphotic zone nitrate (Spearman’s rank correlation; $r_s = 0.99$), twilight zone $\delta^{15}\text{NNO}_3$ ($r_s = 0.94$), twilight zone $\delta^{15}\text{NPOM}$ ($r_s = 0.95$), NPP ($r_s = 0.69$), zooplankton grazing ($r_s = 0.69$) and water column denitrification ($r_s = 0.88$) and sedimentary denitrification ($r_s = 0.90$) seen in the full model (Figure 4). Moderate agreement for nitrogen fixation in both the direct effect ($r_s = 0.65$) and indirect effect simulations ($r_s = 0.61$) suggested that both warming and circulation changes were equally important in determining its full response, but that nitrogen fixation by itself was not sufficient to force the isotopic trends associated with climate change. The effect of climate change on biogeochemical processes via altering substrate availability was therefore a major cause of 21st century increases in nitrogen limitation, resultant NPP declines, shifts in the sources and sinks of nitrogen, and the accompanying trends in nitrogen isotopes that fingerprint the response of the nitrogen cycle.”

New schematic (change is the numbers for aid in explanation):

I'm confused here about the discussion of the drivers of ¹⁵N concentration without consideration of the isotope ratio that should drive the Δ¹⁵N values. In this same section, I would expect that a nutrient redistribution driven by climate change could lead to a redistribution of N isotopes, but it should result in increases in some regions to compensate the decreases in others. Overall, if there is no new N added, an isotope mass balance should be conserved. I think this discussion would benefit from this perspective of a redistribution of N isotopes and discuss where corresponding increases in Δ¹⁵N occur that offset the decreases in the subtropical thermocline.

It is possible that there are alternative contributions such as those raised by the reviewer, such as denitrification, N₂ fixation, zooplankton recycling and the redistribution of isotopes by the circulation. We therefore agree that these alternatives must be addressed, and have addressed them in sections 3 and 4.

Specifically, in regard to the effects of circulation, we feel that the primary role of biogeochemistry is clear from our flux analysis of ¹⁵NO₃ in the twilight zone discussed on lines 176 – 184.

An important addition to section 3 is:

Lines 165 – 168: *“Potential contributors included a decrease in phytoplankton production and fractionation due to nitrogen limitation, an increase in nitrogen fixation, an increase in zooplankton recycling, a decrease in denitrification, or a physical redistribution of the dissolved nitrogen compounds by a changing circulation (or any combination of these processes).”*

Which we then discuss in the paragraphs that follow.

Lines 178-180: I think it would have helped for this description to come earlier. In many of the earlier references to ‘climate change’, I wondered what mechanism was coming into play.

We have made this more explicit in the introduction. It is addressed in our earlier response.

Lines 396-398: What is meant by ‘mean conditions’ in these contexts?

The multi-annual mean over 20 years from 2081-2100 in both preindustrial control and the anthropogenic experiments. This has been made clearer.

Lines 349 – 353: *“Anthropogenic effects to nitrogen cycling were quantified by comparing mean conditions over the last twenty years of the 21st century (2081-2100) with mean conditions over the final twenty years of the preindustrial control simulation, while effects on nitrogen isotopes were quantified by comparing mean conditions over the last twenty years of the 21st century (2081-2100) with mean conditions over the historical period (1986-2005) from the same simulation.”*

Lines 431-432: A little more explanation would have helped in this section. Specifically, what is meant by ‘annually-average values’ and ‘normalized using the linear slope and mean of the preindustrial control experiment’? What features from the preindustrial control time series were ‘centered on zero’?

We have made this explanation clearer and directed the reader to the supplementary Figure 10 where this explanation is made graphically.

Lines 384 – 389: *“ToE was calculated at each grid cell within both the euphotic and twilight zones (depth-averaged) and using annually-average fields of tracers. We therefore ignore temporal trends and variability equal to and finer than seasonal scales. Raw timeseries were first detrended and normalised using the linear slope and mean of the preindustrial control experiment, such that the preindustrial control timeseries was varied about zero while anomalous trends in experiments with climate change and/or nitrogen deposition deviated from zero.”*

Line 393: *“A graphical representation of this process is shown in Supplementary Figure 10.”*

Line 490: What is the origin of the $(1 - N_{lim} * ENPP / 1000)$ term in this equation?

This is how an open-system fraction of nitrogen isotopes by phytoplankton operates (Sigman and Fripiat 2019). The degree of limitation (N_{lim}) is multiplied by the fractionation factor (ϵ_{phy}), which gives a value between 0 and 5 ‰. This is divided by 1000 to remove the ‰ and convert to a fraction, and is then subtracted from 1 to give a number very close to 1. If full fractionation at 5 ‰ occurs

(under N replete conditions), then phytoplankton will take up roughly 0.995 units of ^{15}N per 1 unit of ^{14}N .

Lines 493-500: Why was a quadratic term chosen to describe respiration? Can you please define and give units for the terms in the equations provided here? I don't understand why you can add respiration and mortality in the last equation since they don't appear to have the same units in the prior equations, with mortality linearly dependent on PON and respiration scaling with PON squared.

This is how respiration and mortality of the higher trophic level are commonly parameterised in ocean biogeochemical models and reflect the same way the NEMO-PISCESv2 is parameterised (Aumont et al. 2015). It reflects density-dependent mortality, via both disease and predation.

Lines 505-506: These equations don't appear to have the same units as N_{exported} has an extra PON term in it that does not exist in N_{recycled} .

These terms have the same units. All tracers in this OD model have the same units in mmol of N m^{-3} .

The reviewer is correct regarding Line 461, which is a typo, and should not have the PON term in it. We have corrected this in our revised manuscript.

Supplementary material:

Note 1: It is more typical for a fractionation factor to be represented by alpha (α) a ratio of isotope ratios, rather than epsilon (ϵ), which is generally given in per mil units.

The reviewer is correct in that fractionation factors may be represented by the fractional effects (α), where α equals the fraction of heavy:light isotope in the product over the fraction of heavy:light isotope in the reactant. ϵ is related to α via: $\alpha = (1 - \epsilon/1000)$.

We have corrected our supplementary note 1 to reflect this notation and have also corrected the sign of our fractionation factors.

Reviewers' Comments:

Reviewer #1:

None

Reviewer #2:

Remarks to the Author:

I reviewed an earlier version of this manuscript.

I find that the revisions have significantly improved the manuscript. The results are much better discussed. I also appreciate seeing the spatial distribution (Figure S1), this certainly demonstrates that this is a highly skillful d15N model. The proposed mechanism is much better illustrated and easier to understand, although I still have a few concerns to address.

I'm surprised by the sharp decline in d15NO3 far away from the upwelling area (after the #2 step) in the 0D model illustration (lowest panel in Figure 3d). I am not aware of this behavior occurring in previous applications of a fractionation model driven only by substrate utilization and export. I don't think this affects the relative change to the future scenario which is the main focus of the manuscript, but I still think this deserves an explanation (at least in the supplemental information) since this is quite unexpected to me.

The small twilight zone (TZ)-d15NO3 decline in the equatorial Pacific and Indian Oceans still puzzles me a bit. The spatial euphotic zone (EZ) d15POM trend moving from east to west (Figures 3a) in the equatorial Pacific fits the illustrated 0D mechanism quite well, but there is no strong impact on TZ-d15NO3 in the region, especially in the climate change only scenario. The western equatorial Pacific is mentioned as an emergent area of significant d15N decline (line 139), but I see this decline around ~0.2 per mil to be rather weak compared to other subtropical areas and is even smaller in the climate change only simulation (Figure 2g).

This is briefly explained to occur due to higher NPP upstream in the Southern Ocean (lines 179-182). I think this is a very important and interesting mechanism that needs further explanation (at least one paragraph) in the main text. I find it fascinating that an increase in NPP/utilization in the Southern Ocean and subsequent mode water transport towards the tropics can almost completely compensate the local isotope effect from a large decline in NPP/utilization in the equatorial Pacific and Indian Oceans, if indeed that is what is actually happening in the 3D model.

I am also surprised there is not a significant TZ-d15NO3 in the southern subtropical Indian Ocean around 30deg S similar to the Pacific and Atlantic sectors, perhaps this could be briefly explained in a couple sentences. A reduction in NPP extends below Madagascar to the southern tip of South Africa (Figures 1c and S5e), whereas the TZ-d15NO3 actually increases here (Figure 2a,g). I don't think the upstream increase in NPP explanation fits here because I see higher increases in upstream NPP in the southern subtropical Atlantic (including along the adjacent coastal zones), yet a large TZ-d15NO3 decline still occurs there in the Atlantic. This is a rather minor point but I still think it is worthwhile to investigate these regional differences.

This leads me to believe this proposed mechanism by the 0D model occurs significantly only in certain physical-biogeochemical regimes in the 3D model. I wonder if these nitrogen utilization dynamics operate differently in iron-limited ocean basins. This may explain why the more iron-limited high latitudes and equatorial Pacific do not express this decline in twilight zone d15NO3, just a thought that crossed my mind. I think the subtropical Indian Ocean also receives less atmospheric dust input than the western Pacific and Atlantic, so maybe it is more iron-limited in the model and this mechanism is not expressed as strongly?

The last sentence of the abstract ("Overall, a shift in the global nitrogen source-sink balance towards

accumulation emerges in the 21st century, and is uniquely fingerprinted by $\delta^{15}\text{N}$ declines.”) gives the impression that nitrogen isotopes are fingerprinting the net accumulation, but there is little resemblance between $\delta^{15}\text{N}$ changes and areas of net accumulation (as shown in the response letter). I recommend to rephrase the end of the sentence to something along the lines “... 21st century, and anthropogenically driven upper ocean nitrogen dynamics are uniquely fingerprinted by $\delta^{15}\text{N}$ ”.

As I detailed in my comments above, the last issue in my view that requires a better explanation before publication is the spatial pattern and “hot spots” (or lack thereof in some locations) of the TZ- $\delta^{15}\text{NO}_3$ decline, especially in the climate change only scenario where the mechanism described in the 0D model should be best expressed. After this issue is better addressed and discussed, I would recommend this manuscript for publication.

Overall this is a nice, interesting study that I learned from and enjoyed reading and reviewing!

Best,
Christopher Somes
GEOMAR Helmholtz Centre for Ocean Research Kiel

Reviewer #3:

Remarks to the Author:

I found the paper to be somewhat improved. However, I still struggle to understand the proposed mechanism of N limitation decreasing $\delta^{15}\text{N}$ - NO_3 . I also find some of the equations presented for the zero dimensional model to be poorly described. Thus, I cannot evaluate their validity. As the results of the larger model appear to depend on the validity of the zero order model, I also cannot evaluate the validity of the implementation of isotopes in the larger model.

line 436: what are the units of u_{max} ? T ? The supplementary table reports that T_{growth} is unitless. So, I don't understand how $T * T_{\text{growth}}$ is unitless. Also, shouldn't u_{max} have units of per day? Which term in this equation gives it those units?

Lines 452-454: I questioned these equations in the first round of review, and I still do not see how respiration and mortality can have the same units. Respiration appears to have units of PON^2 per day, and mortality has units of PON per day.

Line 459: Same question about the units of T .

Reviewer 2

I reviewed an earlier version of this manuscript.

I find that the revisions have significantly improved the manuscript. The results are much better discussed. I also appreciate seeing the spatial distribution (Figure S1), this certainly demonstrates that this is a highly skillful $\delta^{15}\text{N}$ model. The proposed mechanism is much better illustrated and easier to understand, although I still have a few concerns to address.

I'm surprised by the sharp decline in $\delta^{15}\text{NO}_3$ far away from the upwelling area (after the #2 step) in the OD model illustration (lowest panel in Figure 3d). I am not aware of this behavior occurring in previous applications of a fractionation model driven only by substrate utilization and export. I don't think this affects the relative change to the future scenario which is the main focus of the manuscript, but I still think this deserves an explanation (at least in the supplemental information) since this is quite unexpected to me.

The small twilight zone (TZ)- $\delta^{15}\text{NO}_3$ decline in the equatorial Pacific and Indian Oceans still puzzles me a bit. The spatial euphotic zone (EZ) $\delta^{15}\text{POM}$ trend moving from east to west (Figures 3a) in the equatorial Pacific fits the illustrated OD mechanism quite well, but there is no strong impact on TZ- $\delta^{15}\text{NO}_3$ in the region, especially in the climate change only scenario. The western equatorial Pacific is mentioned as an emergent area of significant $\delta^{15}\text{N}$ decline (line 139), but I see this decline around ~ 0.2 per mil to be rather weak compared to other subtropical areas and is even smaller in the climate change only simulation (Figure 2g).

This is briefly explained to occur due to higher NPP upstream in the Southern Ocean (lines 179-182). I think this is a very important and interesting mechanism that needs further explanation (at least one paragraph) in the main text. I find it fascinating that an increase in NPP/utilization in the Southern Ocean and subsequent mode water transport towards the tropics can almost completely compensate the local isotope effect from a large decline in NPP/utilization in the equatorial Pacific and Indian Oceans, if indeed that is what is actually happening in the 3D model.

I am also surprised there is not a significant TZ- $\delta^{15}\text{NO}_3$ in the southern subtropical Indian Ocean around 30deg S similar to the Pacific and Atlantic sectors, perhaps this could be briefly explained in a couple sentences. A reduction in NPP extends below Madagascar to the southern tip of South Africa (Figures 1c and S5e), whereas the TZ- $\delta^{15}\text{NO}_3$ actually increases here (Figure 2a,g). I don't think the upstream increase in NPP explanation fits here because I see higher increases in upstream NPP in the southern subtropical Atlantic (including along the adjacent coastal zones), yet a large TZ- $\delta^{15}\text{NO}_3$ decline still occurs there in the Atlantic. This is a rather minor point but I still think it is worthwhile to investigate these regional differences.

This leads me to believe this proposed mechanism by the OD model occurs significantly only in certain physical-biogeochemical regimes in the 3D model. I wonder if these nitrogen utilization dynamics operate differently in iron-limited ocean basins. This may explain why the more iron-limited high latitudes and equatorial Pacific do not express this decline in twilight zone $\delta^{15}\text{NO}_3$, just a thought that crossed my mind. I think the subtropical Indian Ocean also receives less atmospheric dust input than the western Pacific and Atlantic, so maybe it is more iron-limited in the model and this mechanism is not expressed as strongly?

The last sentence of the abstract ("Overall, a shift in the global nitrogen source-sink balance towards accumulation emerges in the 21st century, and is uniquely fingerprinted by $\delta^{15}\text{N}$ declines.") gives

the impression that nitrogen isotopes are fingerprinting the net accumulation, but there is little resemblance between $\delta^{15}\text{N}$ changes and areas of net accumulation (as shown in the response letter). I recommend to rephrase the end of the sentence to something along the lines "... 21st century, and anthropogenically driven upper ocean nitrogen dynamics are uniquely fingerprinted by $\delta^{15}\text{N}$ ".

As I detailed in my comments above, the last issue in my view that requires a better explanation before publication is the spatial pattern and "hot spots" (or lack thereof in some locations) of the TZ- $\delta^{15}\text{NNO}_3$ decline, especially in the climate change only scenario where the mechanism described in the OD model should be best expressed. After this issue is better addressed and discussed, I would recommend this manuscript for publication.

Overall this is a nice, interesting study that I learned from and enjoyed reading and reviewing!

Best,
Christopher Somes
GEOMAR Helmholtz Centre for Ocean Research Kiel

Once again, we want to thank Chris Somes for his positive and useful comments that help us improve the manuscript. We address his comments (in italics) below.

Comment 1:

I'm surprised by the sharp decline in $\delta^{15}\text{NNO}_3$ far away from the upwelling area (after the #2 step) in the OD model illustration (lowest panel in Figure 3d). I am not aware of this behavior occurring in previous applications of a fractionation model driven only by substrate utilization and export. I don't think this affects the relative change to the future scenario which is the main focus of the manuscript, but I still think this deserves an explanation (at least in the supplemental information) since this is quite unexpected to me."

The sharp decline in $\delta^{15}\text{NNO}_3$ that occurs after step 2 in Figure 3d is explained by a combination of increasing nitrogen-limitation and increasing reliance of phytoplankton growth on regenerated nitrogen (i.e. due to recycling). This means that our model does not follow the expectations of a closed system. We have added supplementary text 3 and a supplementary figure (10) to explain these dynamics in more detail.

Supplementary Note 3 reads:

"Our 0-D isotope model contains three pools of nitrogen. These are dissolved inorganic nitrogen (DIN), particulate organic nitrogen (PON) and exported nitrogen (ExpN), all in the same units of mmol N m^{-3} . Initially, the DIN pool represents the bulk of total nitrogen in the model. Over time, phytoplankton consume DIN and increase the pool of PON. This subsequently increases ExpN as the rate of export is to a first order dependent on the biomass of PON (Supplementary Figure 10a).

Rates of DIN uptake [DIN→PON], PON recycling [PON→DIN] and PON export [PON→ExpN] all increase as the PON pool increases (Supplementary Figure 10b). Once DIN is depleted to a limiting concentration near zero ($\sim 0.07 \text{ mmol m}^{-3}$), the PON pool no longer grows. Now, the rate of DIN uptake closely matches the rate of PON recycling, such that all primary production occurs through regeneration. The PON pool begins to decline as nitrogen is lost via export.

The shift into a DIN-limited system, where primary production is regenerated rather than new, also coincides with a decrease in fractionation associated with DIN uptake (Supplementary Figure 10c). Here, the limiting concentrations of DIN mean that phytoplankton have less preference for the lighter isotope over the heavier isotope (Needoba et al., 2004; Karsh et al., 2012).

These dynamics result in the $\delta^{15}\text{N}$ trends shown in Supplementary Figure 10d. High rates of new production and strong fractionation initially drive an increase in $\delta^{15}\text{N}_{\text{DIN}}$, which is paralleled by $\delta^{15}\text{N}_{\text{PON}}$ at a constant offset equal to ϵ_{phy} (~5 ‰). As DIN is consumed towards limiting concentrations, all primary production transitions from mostly new to mostly regenerated (i.e. supported by recycling). New DIN produced through recycling has an isotopic signature equal to $\delta^{15}\text{N}_{\text{PON}}$, and due to the low concentration of remaining DIN, $\delta^{15}\text{N}_{\text{DIN}}$ approximates $\delta^{15}\text{N}_{\text{PON}}$. Thus, phytoplankton under DIN-limited conditions take on an isotopic signature close to that of the PON pool, with a constant offset equal to ϵ_{phy} , which under DIN-limited conditions is near zero. Over the course of the simulation, the combined DIN and PON pools have become increasingly enriched in ^{15}N due to export of PON that is relatively depleted in ^{15}N .

Under a closed system with no recycling, isotopic dynamics instead obey the Rayleigh model (Mariotti et al., 1981; Sigman and Fripiat, 2019). In such a case, the isotopic signature of the reactant ($\delta^{15}\text{N}_{\text{DIN}}$) would increase towards infinity as it is consumed towards zero, while the isotopic signature of the product ($\delta^{15}\text{N}_{\text{PON}}$) would increase towards the initial isotopic signature of the reactant. The dynamics of our simulations differ from this closed-system model because we (i) recycle some of the product back to the reactant pool (i.e. a back-reaction), and (ii) export of some of the product out of the system.

And is supplemented with this figure:

Supplementary Figure 10. Zero-dimensional model following a water parcel within the euphotic zone from point of upwelling in the tropics to nitrogen depletion in the gyre. **a**, Evolution of major pools of dissolved inorganic nitrogen (DIN), particulate organic matter (POM), exported matter (ExpM) and the total concentration of nitrogen, all in units of mmol N m^{-3} . **b**, Rates of DIN uptake, POM recycling and POM export. **c**, Strength of fractionation associated with DIN uptake by phytoplankton. **d**, Accumulated isotopic signatures of the major pools.

Comment 2:

The small twilight zone (TZ)-d15NO₃ decline in the equatorial Pacific and Indian Oceans still puzzles me a bit. The spatial euphotic zone (EZ) d15POM trend moving from east to west (Figures 3a) in the equatorial Pacific fits the illustrated OD mechanism quite well, but there is no strong impact on TZ-d15NO₃ in the region, especially in the climate change only scenario. The western equatorial Pacific is mentioned as an emergent area of significant d15N decline (line 139), but I see this decline around ~0.2 per mil to be rather weak compared to other subtropical areas and is even smaller in the climate change only simulation (Figure 2g).

This is briefly explained to occur due to higher NPP upstream in the Southern Ocean (lines 179-182). I think this is a very important and interesting mechanism that needs further explanation (at least one paragraph) in the main text. I find it fascinating that an increase in NPP/utilization in the Southern Ocean and subsequent mode water transport towards the tropics can almost completely compensate the local isotope effect from a large decline in NPP/utilization in the equatorial Pacific and Indian Oceans, if indeed that is what is actually happening in the 3D model.

I am also surprised there is not a significant TZ-d15NO₃ in the southern subtropical Indian Ocean around 30deg S similar to the Pacific and Atlantic sectors, perhaps this could be briefly explained in a couple sentences. A reduction in NPP extends below Madagascar to the southern tip of South Africa (Figures 1c and S5e), whereas the TZ-d15NO₃ actually increases here (Figure 2a,g). I don't think the upstream increase in NPP explanation fits here because I see higher increases in upstream NPP in the southern subtropical Atlantic (including along the adjacent coastal zones), yet a large TZ-d15NO₃ decline still occurs there in the Atlantic. This is a rather minor point but I still think it is worthwhile to investigate these regional differences.

This leads me to believe this proposed mechanism by the OD model occurs significantly only in certain physical-biogeochemical regimes in the 3D model. I wonder if these nitrogen utilization dynamics operate differently in iron-limited ocean basins. This may explain why the more iron-limited high latitudes and equatorial Pacific do not express this decline in twilight zone d15NO₃, just a thought that crossed my mind. I think the subtropical Indian Ocean also receives less atmospheric dust input than the western Pacific and Atlantic, so maybe it is more iron-limited in the model and this mechanism is not expressed as strongly?

We agree, and have included some small additional explanations in sections 2 and 3 of the results that address and acknowledge these comments: (i) highlighting the weak twilight zone $\delta^{15}\text{N}_{\text{NO}_3}$ declines in the equatorial Pacific and Indian Oceans, and (ii) explaining increased twilight zone $\delta^{15}\text{N}_{\text{NO}_3}$ and/or muted changes in the South Indian and South Pacific Oceans. In doing so, we address the reviewer's final comment, which is (iii) the possibility of different non-local processes affecting the twilight zone $\delta^{15}\text{N}_{\text{NO}_3}$ in other regions.

Section 2 has been altered on lines 135-139:

"In contrast, twilight zone $\delta^{15}\text{N}_{\text{NO}_3}$ declined more uniformly by >0.2‰ across the tropical and subtropical Pacific and the Atlantic by 2081-2100 (Figure 2a), with stronger signals in the gyres where nitrate concentrations are lowest (Supplementary Figure 9). Unlike the euphotic zone, these declines in the twilight zone are within detection limits of observational methods, since concentrations of nitrate exceed the 0.3 mmol m⁻³ threshold required for isotopic measurement⁴³."

and we have added a supplementary figure (9):

Supplementary Figure 9. Average nitrate (NO₃) concentration within the twilight zone from 1851-2100 under the full anthropogenic simulation (climate change and anthropogenic nitrogen deposition).

Section 3 has also been altered, mainly in the second paragraph from lines 178-200, which now reads:

“Firstly, the presence of ¹⁵N-depleted organic matter sinking from the overlying euphotic zone is a first order driver of the ⁵¹⁵N_{NO₃ declines in the twilight zone. Widespread declines in euphotic zone ⁵¹⁵N_{POM} (Figure 3a) thus delivered less ¹⁵N to twilight zones across large parts of the lower latitude ocean following the remineralisation of sinking organic material. This driver was further supported by an offline analysis of purely biogeochemical ¹⁵NO₃ fluxes in our global model. The only internal biogeochemical source of nitrate is from nitrification of the ammonium that forms following remineralisation, while the main sinks are primary production and denitrification. By isolating the fluxes of these individual sources and sinks within each grid cell, we disentangled local biogeochemical effects from circulation effects (see Methods). Biogeochemical fluxes tended to deplete ⁵¹⁵N_{NO₃ across the tropics and subtropics, and the ensuing changes to ⁵¹⁵N_{NO₃ were greater than observed in the full model (Figure 3b). This indicates that physical sources and sinks (i.e. ocean transports) partially compensated for the biogeochemical effects. Smoothing of the strong gradients set by biogeochemical fluxes is consistent with the role of ocean physics for upward mixing of deep nitrate with relatively constant ⁵¹⁵N_{NO₃ values. The only exceptions to this picture were denitrification zones in the North Indian and East Pacific Oceans, where denitrification rates increased and raised ⁵¹⁵N_{NO₃, and in South Indian and South Pacific downstream of Southern Ocean mode waters^{44,45}, where upstream increases in NPP (Figure 1c) raised ⁵¹⁵N_{NO₃. The upstream isotopic enrichment of mode waters exceeded the declines caused by nutrient limitation in the Indian Ocean and partially compensated for the declines caused by nutrient limitation in the Pacific. South Atlantic twilight zones were not affected, as its mode waters form at higher latitudes in the southeast Pacific and on deeper, denser isopycnals⁴⁵, which outcrop poleward of Subantarctic increases in NPP. Despite these regional exceptions, the broad biogeochemical tendency was to lower twilight zone ⁵¹⁵N_{NO₃ and this was linked to depleted ⁵¹⁵N_{POM} sinking out of subtropical euphotic zones.”}}}}}}}

These additions address the comments of the reviewer.

Comment 3:

The last sentence of the abstract (“Overall, a shift in the global nitrogen source-sink balance towards accumulation emerges in the 21st century, and is uniquely fingerprinted by $\delta^{15}\text{N}$ declines.”) gives the impression that nitrogen isotopes are fingerprinting the net accumulation, but there is little resemblance between $d^{15}\text{N}$ changes and areas of net accumulation (as shown in the response letter). I recommend to rephrase the end of the sentence to something along the lines “... 21st century, and anthropogenically driven upper ocean nitrogen dynamics are uniquely fingerprinted by $d^{15}\text{N}$ ”.

We agree with the reviewer and have altered the final sentence of the abstract in line with their suggestion:

“Overall, a shift in the global nitrogen source-sink balance towards accumulation emerges in the 21st century, and the anthropogenically-driven response is uniquely fingerprinted by upper ocean $\delta^{15}\text{N}$ declines.”

Note that it is slightly different from their suggestion so that we remain within the 150 word limit.

Comment 4:

As I detailed in my comments above, the last issue in my view that requires a better explanation before publication is the spatial pattern and “hot spots” (or lack thereof in some locations) of the TZ- $d^{15}\text{NO}_3$ decline, especially in the climate change only scenario where the mechanism described in the OD model should be best expressed. After this issue is better addressed and discussed, I would recommend this manuscript for publication.

We agree and have addressed this concern in the responses and amendments/additions to the text that are detailed above.

References

- Karsh, K. L., Granger, J., Kritee, K., and Sigman, D. M. (2012). Eukaryotic Assimilatory Nitrate Reductase Fractionates N and O Isotopes with a Ratio near Unity. *Environ. Sci. Technol.* 46, 5727–5735. doi:10.1021/es204593q.
- Mariotti, A., Germon, J., Hubert, P., Kaiser, P., Letolle, R., Tardieux, A., et al. (1981). Experimental determination of nitrogen kinetic isotope fractionation: some principles; illustration for the denitrification and nitrification process. *Plant Soil* 62, 413–430. doi:10.1007/BF02374138.
- Needoba, J. A., Sigman, D. M., and Harrison, P. J. (2004). The mechanism of isotope fractionation during algal nitrate assimilation as illuminated by the $^{15}\text{N}/^{14}\text{N}$ of intracellular nitrate. *J. Phycol.* 40, 517–522. doi:10.1111/j.1529-8817.2004.03172.x.
- Sigman, D. M., and Fripiat, F. (2019). “Nitrogen Isotopes in the Ocean,” in *Encyclopedia of Ocean Sciences* (Elsevier), 263–278. doi:10.1016/B978-0-12-409548-9.11605-7.

Reviewer 3:

Reviewer #3 (Remarks to the Author):

I found the paper to be somewhat improved. However, I still struggle to understand the proposed mechanism of N limitation decreasing $\delta^{15}\text{N}$ - NO_3 . I also find some of the equations presented for the zero dimensional model to be poorly described. Thus, I cannot evaluate their validity. As the results of the larger model appear to depend on the validity of the zero order model, I also cannot evaluate the validity of the implementation of isotopes in the larger model.

line 436: what are the units of u_{max} ? T? The supplementary table reports that T_{growth} is unitless. So, I don't understand how $T \cdot T_{\text{growth}}$ is unitless. Also, shouldn't u_{max} have units of per day? Which term in this equation gives it those units?

Lines 452-454: I questioned these equations in the first round of review, and I still do not see how respiration and mortality can have the same units. Respiration appears to have units of PON^2 per day, and mortality has units of PON per day.

Line 459: Same question about the units of T.

We want to thank the reviewer for their useful comments. They have asked for a clarification of some of the equations/units within the 0-D model. We address their comments (in italics) below.

In addition to the specific comments below, we have improved our description of the 0-D model in the methods section, ensuring that all equations and constants are referred to in the text, and by the addition of a Supplementary Note 3 and Supplementary Figure 10, which more completely explains the dynamics.

Comment 1:

line 436: what are the units of u_{max} ? T? The supplementary table reports that T_{growth} is unitless. So, I don't understand how $T \cdot T_{\text{growth}}$ is unitless. Also, shouldn't u_{max} have units of per day? Which term in this equation gives it those units?

μ_{max} is in units of per day (day^{-1}). Temperature (T) is in units of $^{\circ}\text{C}$. T_{growth} is typically defined as unitless, because it is a power scaling constant defined by Eppley (1972). However, according to the the equation $\ln r = 0.6 \text{ day}^{-1} * e^{T \cdot T_h}$, which returns units of day^{-1} , T_{growth} could more accurately be described in units of ($^{\circ}\text{C}^{-1}$). At 18°C , μ_{max} is equal to $\sim 1.9 \text{ day}^{-1}$. These pieces of information have been added to the Supplementary Table, and the line "At a constant temperature of 18°C , $i_{\text{m,r}}$ is equal to $\sim 1.9 \text{ day}^{-1}$." has been added to the methods on line 448.

Comment 2:

Lines 452-454: I questioned these equations in the first round of review, and I still do not see how respiration and mortality can have the same units. Respiration appears to have units of PON^2 per

day, and mortality has units of PON per day.

The reviewer has identified that our units as presented in the Supplementary Table were not correct. The quadratic mortality term is multiplied by a coefficient in units of $(\text{mmol N m}^{-3})^{-1} \text{ day}^{-1}$, while the linear respiration loss term is multiplied by a coefficient in units of day^{-1} . These terms have been updated in the methods and supplementary table.

Comment 3:

Line 459: Same question about the units of T.

Recycling is calculated as a fraction of the total amount of detritus, hence it is unitless until multiplied against this pool. T_{growth} in units of $^{\circ}\text{C}^{-1}$ is multiplied by T in units of $^{\circ}\text{C}$ to give the exponential-scaling that increases the fraction of recycled detritus above its minimum of 0.4.

Reviewers' Comments:

Reviewer #2:

Remarks to the Author:

I reviewed previous versions of this manuscript.

I think the authors have done a nice job with the revisions and I only have one remaining minor point.

After reading the title and abstract only, I still feel readers may think that the isotopes "fingerprint" the source-sink imbalance given how often it is mentioned (lines 2, 20, 24, and 26) in the title and abstract, especially if readers don't fully read the main text where these dynamics are well described.

For example, the ordering of the statements (lines 24-25) "... to drive a global increase in nitrogen sources over sinks. Consequently, ~75% of twilight zones across the low latitudes develop anomalously low d15N by 2060." gives the impression that the source/sink imbalance causes the anomalously low d15N. Maybe the latter sentence would be better positioned before the previous sentence?

I would recommend to at least briefly mention in the abstract the main mechanism driving the intensifying nitrogen limitation and anomalously low d15N, i.e. reduced nutrient supply due to warming-induced stratification. Currently, only "climate change impacts" is mentioned, which is somewhat vague and without direct reference to the isotopes in that sentence on lines 23-24.

I leave it to the authors to implement these final minor revisions in the abstract and recommend the manuscript for publication.

Cheers,

Christopher Someș

GEOMAR Helmholtz Centre for Ocean Research Kiel

Reviewer #3:

Remarks to the Author:

I have read through all of the materials provided, and thank the authors for responding patiently to my naive questions. In addition, I found the review by Chris Someș, and the author's response to that review, informative and reassuring. One of my main sources of confusion has been that if the decrease in twilight zone d15N is predominantly driven by N limitation and removal of 15N POM from the euphotic zone, this 15N-enriched material should show up somewhere. I think this is now better described in the paper. One final suggestion that I think might help clarify the mechanism, is if the increase d15NNO3 that shows up in the twilight zone in figure 3b could be included in the twilight zone layer of figure 3d. In the bottom part of that figure, it appears that through much of regions 1 and 2, the future d15NPOM is higher than historical (that export of high d15NPON), so if I understand correctly, wouldn't it make sense to show the twilight zone d15NNO3 being higher than historical in regions 1 and 2, and lower primarily in region 3? If so, it might make the illustration more intuitive to show that in the twilight zone d15NNO3 layer?

Reviewer #2 (Remarks to the Author):

I reviewed previous versions of this manuscript.

I think the authors have done a nice job with the revisions and I only have one remaining minor point.

After reading the title and abstract only, I still feel readers may think that the isotopes “fingerprint” the source-sink imbalance given how often it is mentioned (lines 2, 20, 24, and 26) in the title and abstract, especially if readers don’t fully read the main text where these dynamics are well described.

For example, the ordering of the statements (lines 24-25) “... to drive a global increase in nitrogen sources over sinks. Consequently, ~75% of twilight zones across the low latitudes develop anomalously low $\delta^{15}\text{N}$ by 2060.” gives the impression that the source/sink imbalance causes the anomalously low $\delta^{15}\text{N}$. Maybe the latter sentence would be better positioned before the previous sentence?

I would recommend to at least briefly mention in the abstract the main mechanism driving the intensifying nitrogen limitation and anomalously low $\delta^{15}\text{N}$, i.e. reduced nutrient supply due to warming-induced stratification. Currently, only “climate change impacts” is mentioned, which is somewhat vague and without direct reference to the isotopes in that sentence on lines 23-24.

I leave it to the authors to implement these final minor revisions in the abstract and recommend the manuscript for publication.

Cheers,
Christopher Somees
GEOMAR Helmholtz Centre for Ocean Research Kiel

We thank Dr. Somees for his constructive critique of our work and have implemented his final suggestions through slight alterations to both the title and abstract. In the abstract, we have altered the placement of ideas so that the declines in $\delta^{15}\text{N}$ cannot be directly attributed to an increase in sources over sinks in the nitrogen budget. This has been done by earlier placing of the sentence with “~75% of the low latitude twilight zone develops anomalously low $\delta^{15}\text{N}$ ”. We mention the important effect of a changing circulation as well. We have also altered the final implication sentence of the abstract, which does not reflect a new conclusion, but simply reflects what we already propose in the discussion: “that $\delta^{15}\text{N}$ changes in the low latitude twilight zone may provide a useful constraint on emerging changes to nitrogen limitation and NPP over the 21st century.”

Any additional edits to the abstract are due to ensuring that the abstract word count remains < 151 words.

The new title now reads:

“Impact of intensifying nitrogen limitation of ocean net primary production is fingerprinted by nitrogen isotopes”

The new abstract now reads:

“The open ocean nitrogen cycle is being altered by increases in anthropogenic atmospheric nitrogen deposition and climate change. How the nitrogen cycle responds will determine long-term trends in net primary production (NPP) in the nitrogen-limited low latitude ocean, but is poorly constrained by uncertainty in how the source-sink balance will evolve. Here we show that intensifying nitrogen limitation of phytoplankton, associated with near-term reductions in NPP, causes detectable declines in nitrogen isotopes ($\delta^{15}\text{N}$) and constitutes the primary perturbation of the 21st century nitrogen cycle. Model experiments show that ~75% of the low latitude twilight zone develops anomalously low $\delta^{15}\text{N}$ by 2060, predominantly due to the effects of climate change that alter ocean circulation, with implications for the nitrogen sources-sink balance. Our results highlight that $\delta^{15}\text{N}$ changes in the low latitude twilight zone may provide a useful constraint on emerging changes to nitrogen limitation and NPP over the 21st century.”

Reviewer #3 (Remarks to the Author):

I have read through all of the materials provided, and thank the authors for responding patiently to my naive questions. In addition, I found the review by Chris Some, and the author's response to that review, informative and reassuring. One of my main sources of confusion has been that if the decrease in twilight zone $\delta^{15}\text{N}$ is predominantly driven by N limitation and removal of ^{15}N POM from the euphotic zone, this ^{15}N -enriched material should show up somewhere. I think this is now better described in the paper. One final suggestion that I think might help clarify the mechanism, is if the increase $\delta^{15}\text{N}_{\text{NO}_3}$ that shows up in the twilight zone in figure 3b could be included in the twilight zone layer of figure 3d. In the bottom part of that figure, it appears that through much of regions 1 and 2, the future $\delta^{15}\text{N}_{\text{POM}}$ is higher than historical (that export of high $\delta^{15}\text{N}_{\text{POM}}$), so if I understand correctly, wouldn't it make sense to show the twilight zone $\delta^{15}\text{N}_{\text{NO}_3}$ being higher than historical in regions 1 and 2, and lower primarily in region 3? If so, it might make the illustration more intuitive to show that in the twilight zone $\delta^{15}\text{N}_{\text{NO}_3}$ layer?

We thank the reviewer for their constructive critique of our work and have implemented their final suggestions by altering panel d of Figure 3.

The Figure has been altered to show where $\delta^{15}\text{N}$ of NO_3 and POM have increased and decreased in the low latitude ocean. The new figure is shown here:

Figure 3. Biogeochemical control on the nitrogen cycle perturbation. **a**, Climate change effect on euphotic zone $\delta^{15}\text{N}_{\text{POM}}$. **b**, Climate change effects on twilight zone $\delta^{15}\text{N}_{\text{NO}_3}$ due to biogeochemical processes only. **c**, Change in euphotic zone bioavailable nitrogen (nitrate plus ammonium). **d**, schematic describing a major mechanism of $\delta^{15}\text{N}$ depletion in our experiments. Less upwelled bioavailable nitrogen (N) within a water parcel travelling from an upwelling zone (1) to the subtropical gyres (3) leads to lower $\delta^{15}\text{N}_{\text{POM}}$ outside of the productive zone. Greater nitrogen limitation of phytoplankton generates less enrichment of ^{15}N in unused nitrogen (1), leading to a lower peak in $\delta^{15}\text{N}_{\text{NO}_3}$ and $\delta^{15}\text{N}_{\text{POM}}$ at the boundary between nitrogen-replete and nitrogen-limited regimes (2). A sharp depletion in $\delta^{15}\text{N}_{\text{NO}_3}$ is recorded in the area where the nitrogen-replete to nitrogen-limited transition was previously located under historical conditions. This depleted signal is carried laterally into the nitrogen-limited gyres (3), and delivered to the twilight zone via

remineralisation of low $\delta^{15}\text{N}_{\text{POM}}$. Future $\delta^{15}\text{N}_{\text{POM}}$ may be higher in regions near to upwelling (equatorial Pacific and Benguela upwelling in **a**) because nitrogen limitation decreases fractionation by phytoplankton, meaning that more ^{15}N is assimilated into organic matter and removed from the euphotic zone.